# MOF-associated complexes ensure stem cell identity and *Xist* repression

**Tomasz Chelmicki[1,2†], Friederike Dündar[1,2,3†], Matthew James Turley[1,2‡], Tasneem Khanam[1‡], Tugce Aktas[1‡], Fidel Ramírez[3], Anne-Valerie Gendrel[4], Patrick Rudolf Wright[5], Pavankumar Videm[5], Rolf Backofen[5,6,7,8], Edith Heard[4], Thomas Manke[3], Asifa Akhtar[1*]**

[1]Department of Chromatin Regulation, Max Planck Institute of Immunobiology and Epigenetics, Freiburg, Germany; [2]Faculty of Biology, University of Freiburg, Freiburg, Germany; [3]Bioinformatics Department, Max Planck Institute for Immunobiology and Epigenetics, Freiburg, Germany; [4]Mammalian Developmental Epigenetics Group, Institute Curie, Paris, France; [5]Bioinformatics Group, Department of Computer Science, University of Freiburg, Freiburg, Germany; [6]BIOSS Center for Biological Signalling Studies, University of Freiburg, Freiburg, Germany; [7]Center for Biological Systems Analysis, University of Freiburg, Freiburg, Germany; [8]Center for Non-Coding RNA in Technology and Health, University of Copenhagen, Frederiksberg, Denmark

*For correspondence: akhtar@ie-freiburg.mpg.de

†These authors contributed equally to this work

‡These authors also contributed equally to this work

**Abstract** Histone acetyl transferases (HATs) play distinct roles in many cellular processes and are frequently misregulated in cancers. Here, we study the regulatory potential of MYST1-(MOF)-containing MSL and NSL complexes in mouse embryonic stem cells (ESCs) and neuronal progenitors. We find that both complexes influence transcription by targeting promoters and TSS-distal enhancers. In contrast to flies, the MSL complex is not exclusively enriched on the X chromosome, yet it is crucial for mammalian X chromosome regulation as it specifically regulates *Tsix*, the major repressor of Xist lncRNA. MSL depletion leads to decreased *Tsix* expression, reduced REX1 recruitment, and consequently, enhanced accumulation of *Xist* and variable numbers of inactivated X chromosomes during early differentiation. The NSL complex provides additional, *Tsix*-independent repression of *Xist* by maintaining pluripotency. MSL and NSL complexes therefore act synergistically by using distinct pathways to ensure a fail-safe mechanism for the repression of X inactivation in ESCs.

## Introduction

Histone acetyl transferases (HATs) are among the key architects of the cellular epigenetic landscape as the acetylation of histones is unanimously associated with transcriptionally active domains. Many HATs also have the ability to acetylate non-histone proteins extending their influence to diverse cellular pathways inside and outside of the nucleus (reviewed in *Sapountzi and Cote, 2011*). Based on their catalytic domains, the HATs are classified into two major families, GCN5 N-acetyl transferases (GNATs) and MYST HATs (named after the founding members MOZ, Ybf2/Sas3, Sas2, Tip60), that encompass diverse sets of protein complexes. The individual complex members enhance and modulate the enzymes' activities, guiding the versatile HATs towards specific functions. GCN5, for example, is part of SAGA, ATAC, and SLIK complexes that are associated with distinct histone tail modifications and differential gene regulation (reviewed in *Lee and Workman, 2007*; *Nagy et al., 2010*). In contrast, one of the well-known members of the MYST family, MOF (also known as: KAT8, MYST1), is rather substrate-specific for lysine 16 of histone H4 (H4K16) (*Akhtar and Becker, 2000*) and its interaction partners are thought to mainly alter the specificity and extent of MOF's H4K16 acetylation (H4K16ac). As part of the male-specific lethal (MSL) complex (MSL1, MSL2, MSL3, MOF, MLE, roX1 and roX2

**eLife digest** Gene expression is controlled by a complicated network of mechanisms involving a wide range of enzymes and protein complexes. Many of these mechanisms are identical in males and females, but some are not. Female mammals, for example, carry two X chromosomes, whereas males have one X and one Y chromosome. Since the two X chromosomes in females contain essentially the same set of genes, one of them undergoes silencing to prevent the overproduction of certain proteins. This process, which is called X-inactivation, occurs during different stages of development and it must be tightly controlled.

An enzyme called MOF was originally found in flies in two distinct complexes—the male-specific lethal (MSL) complex, which forms only in males, and the non-specific lethal (NSL) complex, which is ubiquitous in both males and females. These complexes are evolutionary conserved and are also found in mammals. While mammalian MOF is reasonably well understood, the MSL and NSL complexes are not, so Chelmicki, Dündar et al. have used various sequencing techniques, in combination with biochemical experiments, to investigate their roles in embryonic stem cells and neuronal progenitor cells in mice.

These experiments show that MSL and NSL complexes engage in the regulation of thousands of genes. Although the two complexes often show different gene preferences, they often regulate the same cellular processes. The MSL/NSL-dependent regulation of X chromosome inactivation is a prime example of this phenomenon.

The MSL complex reduces the production of an RNA molecule called Xist, which is responsible for the inactivation of one of the two X chromosomes in females. The NSL complex, meanwhile, ensures the production of multiple proteins that are crucial for the development of embryonic stem cells, and are also involved in the repression of X inactivation.

This analysis sheds light on how different complexes can cooperate and complement each other in order to reach the same goal in the cell. The knowledge gained from this study will pave the way towards better understanding of complex processes such as embryonic development, organogenesis and the pathogenesis of disorders like cancer.

lncRNAs) in *Drosophila melanogaster*, MOF is recruited to the single X chromosome of male flies. The subsequent spreading of H4K16 acetylation results in transcriptional upregulation of the male X chromosome, the major means of *D. melanogaster* dosage compensation (reviewed in *Conrad and Akhtar, 2011*). In addition to the highly specialized MSL-associated role, MOF is also involved in the more universal and sex-independent regulation of housekeeping genes within the non-specific lethal (NSL) complex (NSL1, NSL2, NSL3, MBD-R2, MCRS2, MOF, WDS) (*Mendjan et al., 2006*; *Raja et al., 2010*; *Feller et al., 2012*; *Lam et al., 2012*).

MOF and most of its interaction partners are conserved in mammals, where MOF is also responsible for the majority of H4K16 acetylation (*Smith et al., 2005*; *Taipale et al., 2005*). MOF is essential for mammalian embryonic development and unlike the male-specific lethality in *Drosophila*, deletion of *Mof* in mice is lethal for both sexes (*Gupta et al., 2008*; *Thomas et al., 2008*). More specifically, mammalian MOF is critical for physiological nuclear architecture (*Thomas et al., 2008*), DNA damage repair (*Gupta et al., 2008*), maintenance of stem cell pluripotency (*Li et al., 2012*), differentiation of T cells (*Gupta et al., 2013*), and survival of post-mitotic Purkinje cells (*Kumar et al., 2011*). Compared to MOF, mammalian MSL and NSL complex members are poorly understood. Nevertheless, the individual complex members appear to have important functions in vivo as mutations of the NSL complex member KANSL1 cause the core phenotype of the 17q21.31 microdeletion syndrome (*Koolen et al., 2012*; *Zollino et al., 2012*) and are common amongst patients with both Down syndrome and myeloid leukemia (*Yoshida et al., 2013*). Another NSL-associated protein, PHF20 has been shown to associate with methylated Lys370 and Lys382 of p53 (*Cui et al., 2012*) and to be required for somatic cell reprogramming (*Zhao et al., 2013a*). WDR5 was shown to be an essential regulator of the core transcription network in embryonic stem cells (*Ang et al., 2011*). The mammalian counterpart of *Drosophila* MSL2 was shown to have the capacity to ubiquitylate p53 (*Kruse and Gu, 2009*) and lysine 34 of histone 2B (*Wu et al., 2011*).

In the study presented here, we set out to dissect the mammalian MOF functions within the MSL and NSL complexes using genome-wide chromatin immunoprecipitation and transcriptome profiles

and biochemical experiments for the core members of MSL and NSL complexes in mouse embryonic stem cells (ESCs) and neuronal progenitor cells (NPCs). We found that the MSL and NSL members possess concurrent, as well as independent functions and that effects generally attributed to MOF are frequently accompanied by the NSL complex. The NSL complex abundantly binds to promoters of broadly expressed genes in ESCs and NPCs. These genes are predominantly downregulated upon depletion of either MOF or KANSL3. In contrast, the MSL complex shows more restricted binding in ESCs, which expands after differentiation, particularly at NPC-specific genes. In addition to promoter-proximal binding, we discover several thousand binding sites of KANSL3 and MSL2 at promoter-distal loci with enhancer-specific epigenetic signatures. The majority of these distal regulatory sites are bound in ESCs, but not in differentiated cells, and genes that are predicted to be targeted by TSS-distal binding of MSL2 are frequently downregulated in sh*Msl2*-treated cells. The distinct, yet synergistic actions of both complexes become very apparent at the X inactivation center (XIC) that encodes numerous non-coding RNAs involved in the silencing of one of the two X chromosomes in differentiating female cells. We show that the MSL but not the NSL complex directly promotes expression of *Tsix*, the inverse transcript and the key murine repressor of *Xist* during early differentiation. Depletion of MSL proteins results in attenuation of *Tsix* transcription, enhanced *Xist* RNA accumulation and 'chaotic' inactivation of variable numbers of X chromosomes during early differentiation. In addition to the very specific effect of MSL1/MSL2-depletion on the XIC genes, we show that MOF together with the NSL complex also influences *Xist* levels, but instead of affecting *Tsix*, MOF and KANSL3 depletion diminish key pluripotency factors involved in repressing *Xist*. Our study provides novel insights into the intricate interplay between MSL and NSL complexes in orchestrating gene expression. Furthermore, we demonstrate how MSLs and NSLs ensure the active state of two X chromosomes in mouse embryonic stem cells via distinct mechanisms.

## Results

### MOF and its complexes show distinct chromatin binding dynamics during differentiation

To examine the behavior of MSL and NSL proteins in a cell type-specific manner, we derived homogeneous populations of multipotent neuronal progenitor cells (NPCs) from mouse embryonic stem cells (ESCs) (*Conti et al., 2005*; *Splinter et al., 2011*; *Gendrel et al., 2014*). We followed the progress of the differentiation process by monitoring cell morphology (*Figure 1A*), as well as protein (*Figure 1B*) and transcript levels of ESC- and NPC-specific markers (*Figure 1—figure supplement 1A–C*). To gain a better understanding of how MOF-associated complexes behave throughout the differentiation process, in parallel to cell type-specific markers, we also monitored the RNA and protein levels of MOF, MSL (MSL1, MSL2), and NSL (KANSL1, KANSL3, MCRS1) complex members (*Figure 1B*, *Figure 1—figure supplement 1A*). Interestingly, MSL and NSL complex members showed distinct RNA and protein dynamics during the process of differentiation: KANSL1 and KANSL3 protein levels remained unchanged, whereas MSL1, MSL2 and MOF became more abundant in NPCs accompanied by increased H4K16 acetylation (H4K16ac) (*Figure 1B*). These results were confirmed using another ES cell line and its NPC derivative (*Figure 1—figure supplement 1D*). The specificities of the antibodies were confirmed by co-immunoprecipitation assays (*Figure 1—figure supplement 2A–C*), as well as shRNA-mediated knockdowns followed by western blot analyses (for individual knockdowns please see below).

To assess the distinct behaviors of the complexes in more detail, we generated genome-wide chromatin binding profiles for MSL1, MSL2 (MSL complex), KANSL3, MCRS1 (NSL complex), and MOF (MSL and NSL). ChIP-seq experiments in ESCs and NPCs (*Figure 2*) yielded large numbers of high-quality DNA sequence reads and excellent agreements between the biological replicates (*Figure 2—figure supplement 1A*, *Supplementary file 1A*). Using MACS for peak calling (*Zhang et al., 2008*) and additional stringent filtering ('Materials and methods'), we scored between 1500 and 15,000 regions of significant enrichments for the different proteins (*Supplementary file 1B*).

To uncover patterns of co-occurrence and independent binding, we used unsupervised clustering on the input-normalized signals. This unbiased approach allowed us to determine five main groups of binding distinguished by different combinations of the proteins and cell-type-specific dynamics. As shown in *Figure 2*, three large clusters of binding sites encompassed regions, where at least 1 of the investigated proteins was present both in ESCs and NPCs (clusters A, B and C). The binding sites of clusters A and B predominantly overlapped with annotated transcription start sites (TSS) in contrast

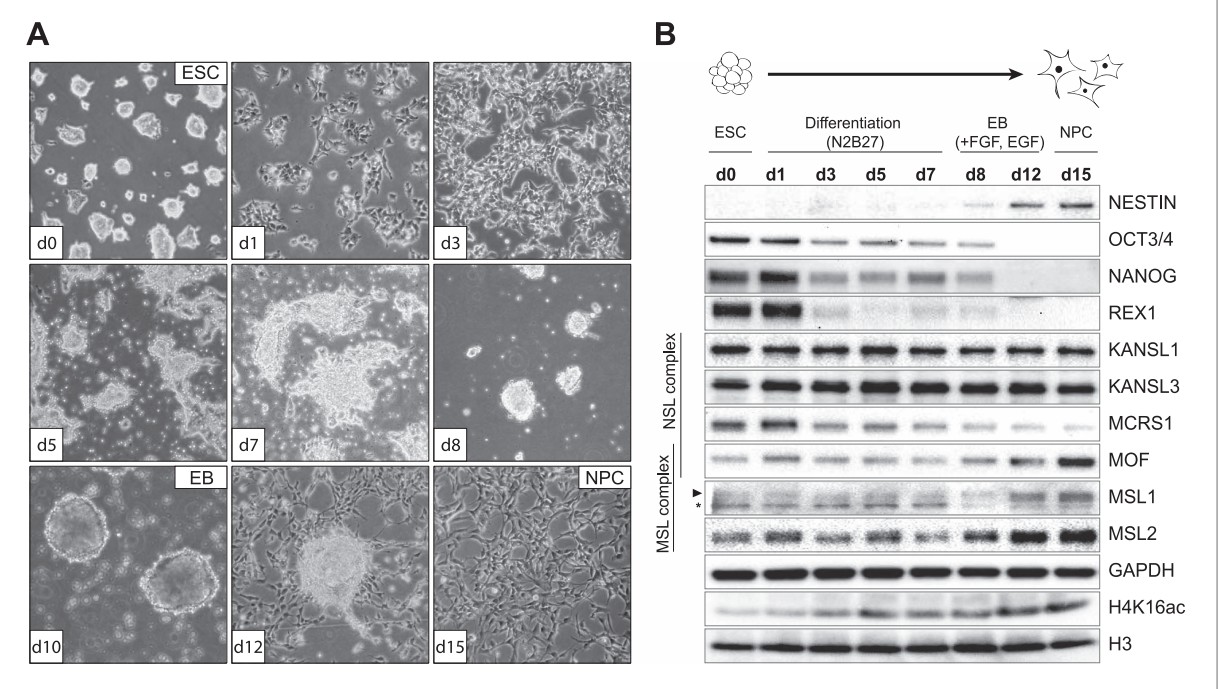

**Figure 1**. Distinct dynamics of MOF, MSL and NSL complexes during differentiation from ESCs to NPCs. (**A**) We monitored the cell morphology during differentiation of mouse embryonic stem cells into neuronal progenitor cells (NPC) via embryoid body formation (EB) with bright field microscopy. The day of differentiation is indicated in white boxes. (**B**) Western blot analysis for ESC to NPC differentiation. Stages of differentiation together with the day of differentiation (d0–d15) are indicated on top. GAPDH and histone 3 (H3) were used as loading controls. For expression analysis see **Figure 1—figure supplement 1**.

The following figure supplements are available for figure 1:

**Figure supplement 1**. Monitoring RNA and protein levels in ESCs and NPCs.

**Figure supplement 2**. Verification of antibodies used in this study.

to the regions that were bound exclusively in ESCs, which tended to contain inter- and intragenic regions (clusters D and E, *Figure 2*). The width of the enrichments did not differ profoundly between the groups (cluster E: 836 bp median width, cluster A: 1782 bp median width). We found surprisingly few regions where MOF associated primarily with MSL complex members. Instead, approximately 80% of all MOF peaks displayed strong KANSL3 and MCRS1 signals (cluster B, see *Figure 2* and *Figure 2—figure supplement 1B*), suggesting a predominant role of the NSL complex among MOF-associated complexes and a more specific role for the MSL complex at subsets of promoters and numerous intergenic and intronic regions. As the different clusters showed distinct enrichment patterns and diverse genomic localization, we set out to analyze the individual groups of binding in more detail.

## The MSL and NSL complexes co-occur on active promoters of constitutively expressed genes in ESCs and NPCs

We first focused on the characterization of target promoters as the majority of MOF-binding was found around the TSS (mostly clusters A and B in *Figure 2*, *Figure 3A*). We identified 8947 TSSs overlapping with ChIP-seq peaks of KANSL3 and/or MCRS1 in ESCs that encompassed virtually all MOF- and MSL-bound TSSs (*Figure 3B*). This pattern did not change substantially in NPCs where TSSs overlapping with MOF peaks almost always (99%) showed significant enrichments of KANSL3 and in 35% of the cases additionally contained a peak of MSL2 (*Figure 3B*, middle panel). Genes that were TSS-bound in ESCs tended to be bound in NPCs as well (*Figure 3B*, middle panel and *Figure 3—figure supplement 1A*). We next generated RNA-seq data for ESCs and NPCs, determined genes that were expressed in both cell types (FPKM >4) and found that all ChIPed proteins preferably bound to

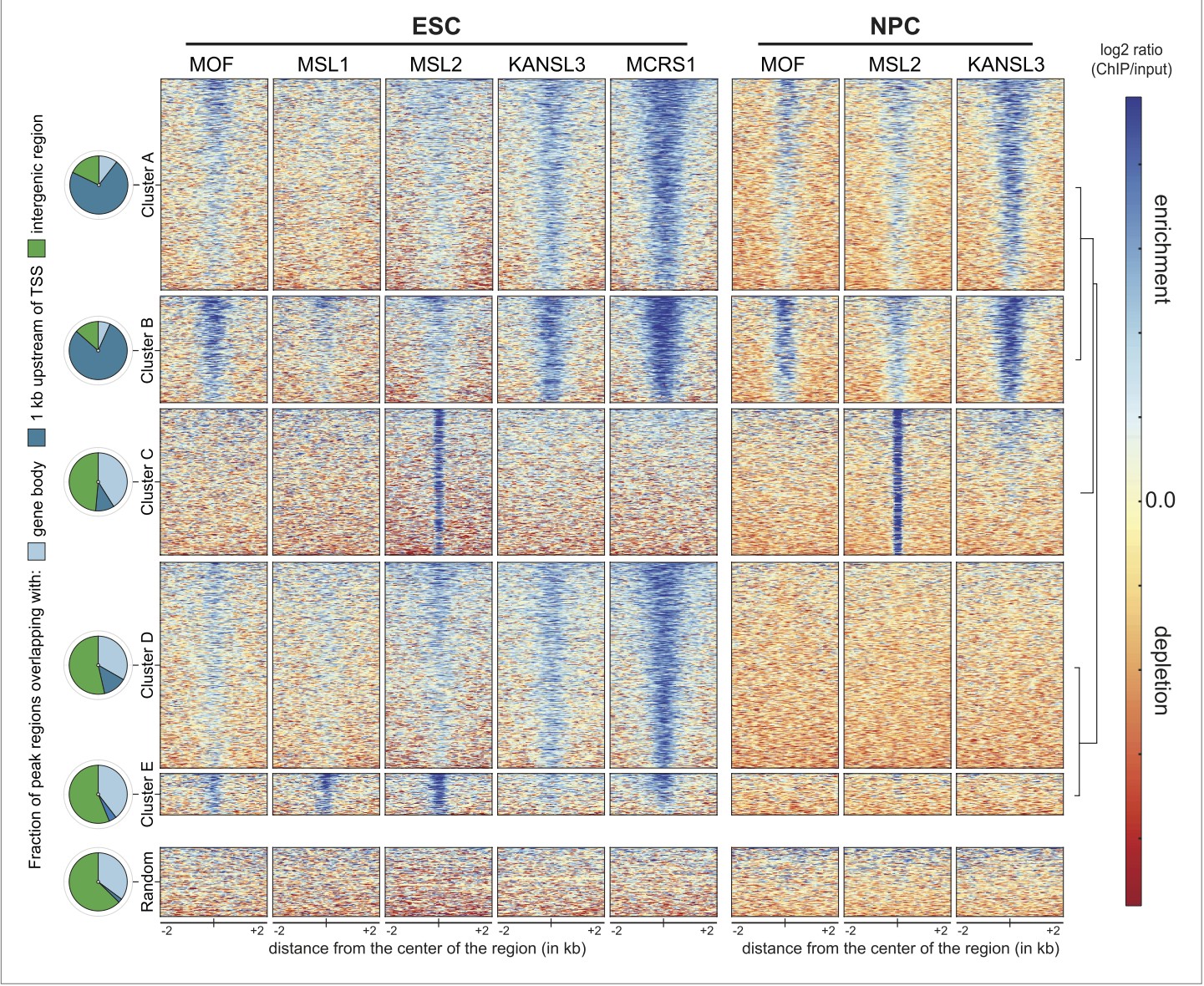

**Figure 2**. Distinct and shared binding sites of MOF and its complexes in mouse ESCs and NPCs. We applied unsupervised clustering on the union of peaks from all ChIP-seq samples and thereby identified five distinct groups of binding for MOF, MSL and NSL proteins in ESCs and NPCs. Shown here are the input-normalized ChIP signals for each cluster of peaks including a size-matched control set of random genomic regions. The order of the regions is the same for all columns. The pie charts on the left indicate the number of regions from each cluster that overlap with gene bodies, the region 1 kb upstream of genes' TSS or intergenic regions.

The following figure supplements are available for figure 2:

**Figure supplement 1**. ChIP-seq quality measures.

the promoters of active genes (**Figure 3C**). Interestingly, in ESCs, genes whose TSSs were bound by members of both complexes showed higher median expression values than genes bound by only one complex (**Figure 3—figure supplement 1B**). In contrast to the differing expression values, analysis of gene ontology (GO) using DAVID (**Huang da et al., 2009**) revealed basic housekeeping functions for both gene groups, regardless of whether they were bound by the NSL complex only or by both MOF-complexes together (**Figure 3—figure supplement 1C**). Consistently, the promoters of all target gene groups were enriched for motifs associated with broad, non-cell-type-specific expression such as ELK1, YY1, CREB, and E2F (**Xie et al., 2005**; **Farre et al., 2007**) and showed profound enrichments of

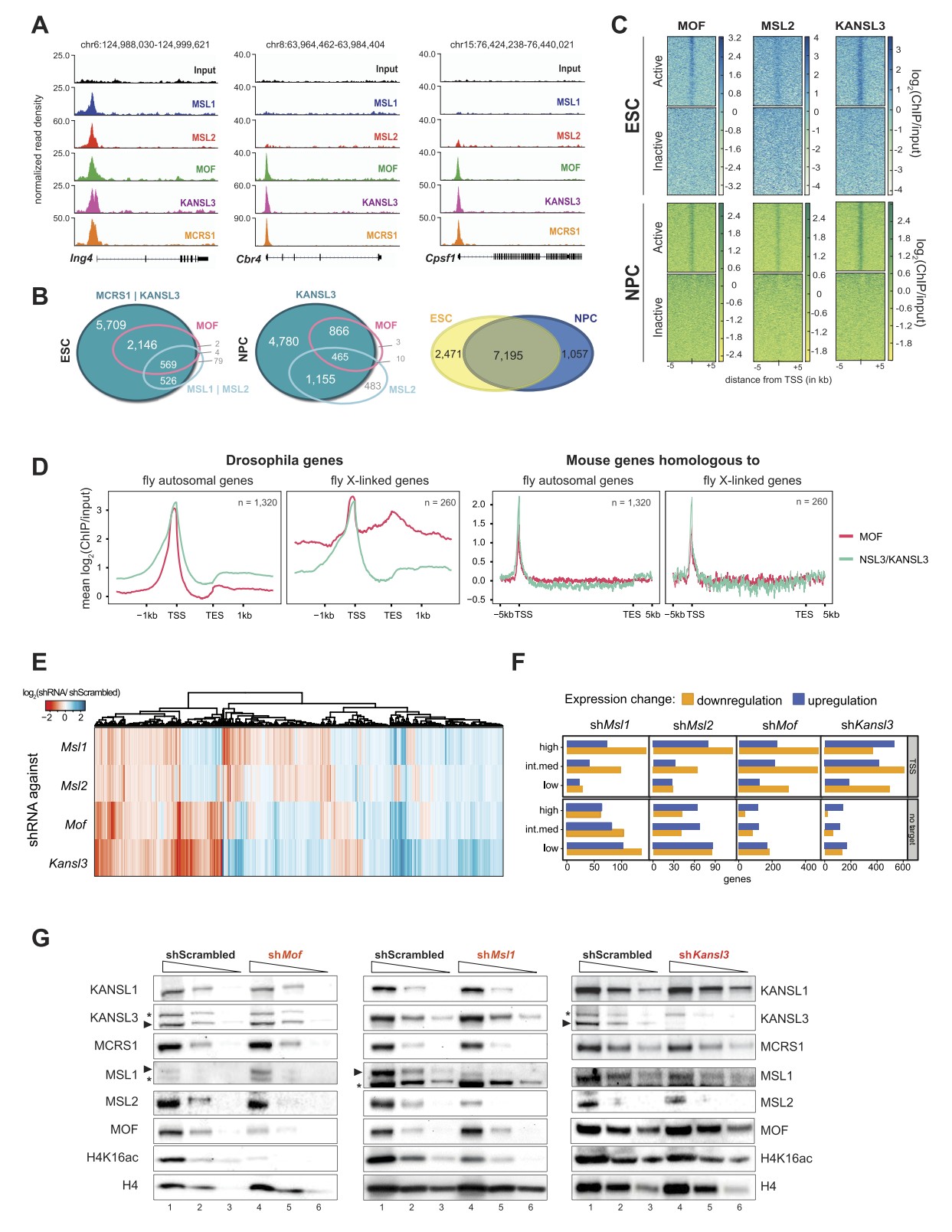

**Figure 3**. Both MOF-complexes bind to the TSS of broadly expressed genes in mouse ESCs and NPCs. (**A**) Genome browser snapshots of genes targeted by MSL and NSL complexes or by the NSL complex only. Signals were sequencing-depth-normalized and from ESCs. For ChIP-qPCR-based validation of the signals see *Figure 3—figure supplement 4B*. (**B**) Venn diagrams of genes whose promoter regions (TSS ± 500 bp) overlapped with
*Figure 3. Continued on next page*

*Figure 3. Continued*

ChIP-seq peaks of NSL complex members (KANSL3 and/or MCRS1), MOF and MSL complex members (MSL1 and/or MSL2). The right-most panel depicts the overlap of genes bound by at least one factor in ESCs and NPCs. (**C**) The heatmaps display the input-normalized ChIP enrichments of MOF, MSL2 and KANSL3 around the TSS of genes that were active in ESCs as well as NPCs based on RNA-seq data that we generated for both cell types. (**D**) Summary plots of genes bound by the NSL complex in *D. melanogaster* for which mouse homologues were found. The input-normalized ChIP-seq signals around the TSS reveal markedly increased binding of MOF for male X-linked fly genes (left panels) that was not recapitulated in the mouse (right panels; ChIP-seq signals from ESCs). Fly genes were scaled to 1.2 kb and values were extracted from published data sets, mouse genes were scaled to 30 kb. (**E**) Heatmap depicting results of RNA-seq experiments from different shRNA-treated cells. The colors correspond to $\log_2$ fold changes (shRNA-treated cells/scrambled control) for genes whose expression was significantly affected in all knockdown conditions. Values were ordered using hierarchical clustering. (**F**) Bar plot of gene counts for different gene classes. We determined significantly up- and downregulated genes for each knockdown condition and binned them according to their expression strength in wild-type ESCs (high, intermediate, low). Then, for each gene, information about the TSS-targeting was extracted from the corresponding ChIP-seq sample. Non-target genes are neither bound at the promoter nor the gene body and were not predicted to be regulated via TSS-distal binding sites in any of the 5 ChIP-seq ESC samples. For details on the target classification see 'Materials and methods'. (**G**) Western blot analysis of MSL and NSL complex members and H4K16 acetylation in scrambled-, *Mof-*, *Msl1-*, and *Kansl3-* shRNA-treated male ESCs. Three concentrations (100%, 30%, 10%) of RIPA extract were loaded per sample. Asterisks mark the position of unspecific bands; triangles indicate the protein of interest.

The following figure supplements are available for figure 3:

**Figure supplement 1**. MSL and NSL complexes target promoters of broadly expressed genes in ESCs and NPCs.

**Figure supplement 2**. The NSL-, but not the MSL-binding mode of *D. melanogaster* is present in mammalian cells.

**Figure supplement 3**. Effects of shRNA-mediated depletion of MOF, MSL1, MSL2 and KANSL3.

**Figure supplement 4**. Assessment of ChIP signals around the TSSs of putative target genes as determined by ChIP-seq.

CpG islands (*Figure 3—figure supplement 1D*), which is indicative of housekeeping genes (*Landolin et al., 2010*). Interestingly, when we analyzed the subset of genes that gained binding of either KANSL3 or MSL2 in NPCs, we found strong enrichments of GO terms related to embryonic development for KANSL3 targets and cell migration and neuronal development for MSL2 targets.

## The TSS-binding of the mouse NSL complex resembles that of the NSL complex in *D. melanogster*

MOF has traditionally been associated with a widespread enrichment along male X-linked genes in flies that is dependent on the MSL proteins (*Figure 3D*, *Figure 3—figure supplement 2A*). In our mammalian profiles, despite the presence of the MSLs, we could neither detect X-specific enrichments of MOF nor broad domains of binding along gene bodies. Furthermore, promoter-distal binding sites consisted of narrow peaks and no evidence of spreading from intronic or intergenic regions was observed (*Figures 2, 3A,D*).

We then examined whether there was a correlation between NSL complex binding in *D. melanogaster* and mouse cells. Indeed, we found that mouse genes that were homologous to NSL complex targets in *D. melanogaster* had a high probability of being bound by the murine NSL complex as well (Pearson's Chi squared test of independence between NSL binding in the fly and the mouse, p-value <2.2e−16). We additionally observed that mouse genes expressed in ESCs and NPCs, whose fly homologues were NSL targets, showed stronger signals for H3K4me3, MOF, KANSL3, and MCRS1 (but not for MSL1 or MSL2) than the mouse homologues of non-NSL-bound *D. melanogaster* genes (*Figure 3—figure supplement 2B*; lists of NSL-bound and NSL-non-bound fly genes were from *Lam et al., 2012*). These findings support the notion that the function in housekeeping gene regulation by the *D. melanogaster* NSL complex is evolutionary conserved.

## Depletion of MSL and NSL complex members results in genome-wide downregulation of TSS-target genes

To dissect the biological consequences of the gene targeting by the different MSL and NSL proteins in ESCs, we systematically depleted core members of both complexes (MOF, KANSL3, MSL1, MSL2) (*Figure 3—figure supplement 3A*). Interestingly, MOF- or KANSL3-depleted cells showed more

severe proliferation defects than MSL1- and MSL2-depleted cells (*Figure 3—figure supplement 3B*). We subsequently performed RNA-seq experiments from shRNA-treated cells and determined their differential expression against the scrambled control to dissect transcriptional outcomes of the depletions at a global level. We found a striking overlap between the differential expression of MSL1 and MSL2 knockdowns and a higher resemblance of MOF-dependent differential expression to that of KANSL3-depletion (*Figure 3E*). When we specifically focused on genes that we had identified as TSS-bound in our ChIP-seq samples, we found that their transcripts tended to be downregulated in all four knockdowns in comparison to untargeted genes which showed higher fractions of upregulation. These effects were independent of the wild-type expression status of the gene or the chromosome (*Figure 3F*, *Figure 3—figure supplement 3C*).

## TSS-binding of MSL1 and KANSL3 does not require MOF

Turning to the assessment of protein levels in shRNA-treated cells, we detected markedly reduced bulk H4K16 acetylation in MSL1- and MOF-depleted cells and only slight reduction upon KANSL3-depletion. This is consistent with previous reports that indicate MSL1 as the major enhancer of MOF's H4K16 acetylation (*Kadlec et al., 2011*) and demonstrate relaxed substrate specificity for the NSL complex (*Zhao et al., 2013b*). In addition, we found that MSL1-depletion affected the levels of MSL2 but not of NSL complex members while the depletion of KANSL3 moderately decreased protein levels for both complexes (*Figure 3G*). ChIP-qPCR assays in MOF-depleted cells revealed that MSL1 and KANSL3 do not require the presence of MOF to bind to gene pro-moters, which is in agreement with previous observations in *D. melanogaster* (*Hallacli et al., 2012*; *Figure 3—figure supplement 4C*).

In summary, our TSS-focused analysis shows that the localized binding of the NSL complex to the promoters of housekeeping genes appears to be a conserved feature between the mammalian and *Drosophila* systems. Unlike in the fly, we do not detect an MSL- and X-chromosome-specific binding mode of MOF in the mouse cell lines. Instead, both complexes narrowly bind to TSSs where their co-occurrence is associated with significantly higher median expression values than those solely bound by the NSL complex. Moreover, we found that MOF is dispensable for the TSS recruitment of its interaction partners and that depletions of the individual proteins predominantly result in the downregulation of TSS-bound genes, further supporting the fact that the promoter-binding of the MSL and the NSL complex is associated with active transcription.

## MSL and NSL complex members individually bind to active enhancers in ESCs

In addition to promoter-proximal binding, where both the MSL and NSL complex tend to (co-)occur constitutively in ESCs and NPCs, we identified a large proportion of binding sites where the proteins were present in a dynamic fashion, that is their binding was observed only in ESCs but not in NPCs (*Figure 2*, clusters D and E). In contrast to the binding mode represented by clusters A and B (*Figure 2*), here MSL2, MCRS1, and KANSL3 were predominantly enriched within introns and intergenic regions that underwent significant CpG methylation upon differentiation (e.g., from median 50% CpG methyl-ation in ESCs to more than 80% in NPCs for cluster D; bisulfite sequencing data from *Stadler et al., 2011*). As shown in *Figure 4A*, CpG methylation in NPCs was particularly pronounced around the center of the regions with significant ChIP enrichments in ESCs, indicating a correlation between the loss of ChIP-seq signal for MOF, MSL1, MSL2, KANSL3 and MCRS1, and DNA methylation upon differ-entiation. In addition, the regions of cluster D and, to a lesser extent the MSL1-rich cluster E (*Figure 2*), showed highly localized enrichments of DNase hypersensitivity sites (DNase HS), RNA Polymerase II (Pol II), p300, methylation of histone 3 on lysine 4 (H3K4me1), and acetylation of histone 3 on lysine 27 (H3K27ac) in ESCs (*Figure 4A*), which are characteristic features of enhancer regions. We thus examined whether MOF and its interaction partners were enriched on known enhancer regions, using lists of typical and super enhancers defined by binding sites of the pluripotency factors SOX2, NANOG, and OCT4 (*Whyte et al., 2013*), as well as sets of active and poised enhancers based on histone mark signatures (*Creyghton et al., 2010*).

Interestingly, MSL2, KANSL3 and MCRS1, but not MOF and MSL1, showed profound enrichments for active and poised ESC enhancers (*Figure 4—figure supplement 1A*) as well as along the regions of super enhancers that have been described as being particularly important for maintenance of cell identity (*Whyte et al., 2013*).

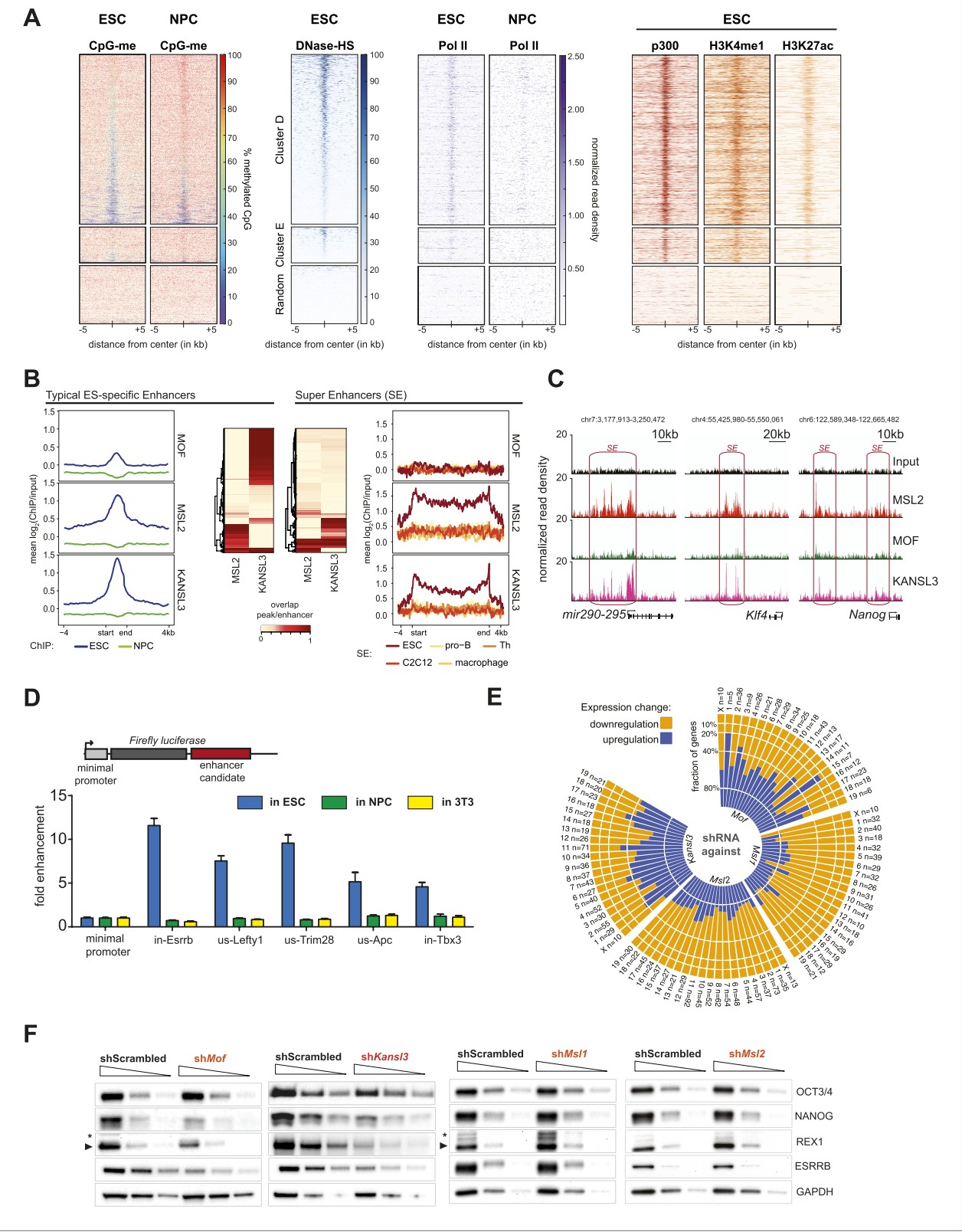

**Figure 4**. MSL and NSL complex members are enriched at regions with enhancer marks in ESCs. (**A**) Shown here are the fractions of methylated cytosines and ChIP-seq read densities of enhancer markers for regions of ESC-specific enrichments of our proteins of interest. We downloaded the different data from public repositories (see ***Supplementary file 3A*** for details) and calculated the values for the regions of the ESC-specific clusters **D** and **E** and
*Figure 4. Continued on next page*

*Figure 4. Continued*

random genomic loci. Most data sets used here were from mouse ESC except one RNA Polymerase II (Pol II) sample from NPC. All heatmaps were sorted according to the DNase hypersensitivity values except for CpG methylation heatmaps which were sorted according to their own values. (**B**) Summary plots of input-normalized ChIP-seq signals along typical (TE) and super enhancers (SE) (*Whyte et al., 2013*). Note that we show the ESC-specific TE only while on the right-hand side we show the signal for SE regions from several cell types. Enhancer regions were scaled to 30 kb (SE) and circa 700 bp (TE). The heatmaps between the summary plots depict how much of each enhancer region overlaps with ChIP-seq peaks of MSL2 or KANSL3. ESC = embryonic stem cells (n = 232), pro-B = progenitor B cells (n = 396), Th = T helper cells (n = 437), C2C12 = myotube cells (n = 536). (**C**) Exemplary genome browser snapshots of annotated super enhancers (SE, pink boxes) for three pluripotency factors displaying the sequencing-depth normalized ESC ChIP-seq signals of MSL2, MOF and KANSL3. See *Figure 4—figure supplement 4C* for additional examples. (**D**) Luciferase assays demonstrate the biological activity of regions bound by MOF-associated proteins in ESCs ('in' stands for intronic region, 'us' indicates that the cloned region is upstream of the gene). The firefly luciferase gene was cloned under a minimal promoter together with the putative enhancer region in ESCs, NPCs, and 3T3 cells. The graphs represent at least three independent experiments performed in technical triplicates; error bars represent SEM. (**E**) Bar plots depicting the fraction of significantly up- and downregulated genes per chromosome in the different shRNA-treated cells compared to shScrambled controls (total number of significantly affected genes per sample and chromosome labels are indicated). All genes counted here were classified as TSS-distal target genes in the respective ChIP-seq experiments. See 'Materials and methods' for details of the classifications. (**F**) Western blot analyses of the pluripotency factors in scrambled-, *Mof*-, *Kansl3*-, *Msl1*-, and *Msl2*-shRNA-treated male ESCs. For additional analyses in female ESCs see *Figure 6C*. The respective dilution (100%, 30%, 10%) of loaded RIPA extract is indicated above each panel. Asterisks mark the position of unspecific bands; triangles indicate the protein of interest. GAPDH was used as the loading control. For antibodies see 'Materials and methods'.

The following figure supplements are available for figure 4:

**Figure supplement 1**. MSL2 and KANSL3 show strong enrichments at typical and super enhancers in ESCs.

**Figure supplement 2**. MOF is moderately enriched at non-canonical enhancers.

**Figure supplement 3**. MSL2 has intergenic binding sites in DNA-hypomethylated regions that are enriched for SMAD3 binding sites.

**Figure supplement 4**. Biological significance of the TSS-distal binding sites of the investigated proteins.

---

The signals of MSL2 and KANSL3 were specific for ESC enhancers and wide-spread along super enhancer regions (*Figure 4B,C*). We noted that enhancers overlapping with MSL2 ChIP-seq peaks tended to show lower KANSL3 enrichments and vice versa, implying that MSL2 and KANSL3 preferred different enhancer regions (heatmaps in *Figure 4B*, *Figure 4—figure supplement 1B*). MOF was not enriched at super enhancers and generally, its binding to TSS-distal sites was much less pronounced than to gene promoters (*Figure 4—figure supplement 1A, 1C*, *Figure 2*). Like for TSS-specific binding, MOF was not alone (87% of TSS-distal MOF peak regions overlapped with either KANSL3 or MSL2). Since a recent report showed H4K16 acetylation to be present at p300- and H3K27-acetylation-independent enhancer regions (*Taylor et al., 2013*), we analyzed the moderate TSS-distal enrichments of MOF in more detail and observed a slight preference for TSS-distal regions that were not overlapping with previously published ESC enhancer regions (*Figure 4—figure supplement 2A*). In fact, we detected the strongest MOF signals in regions with rather low enrichments of known enhancer marks (see DNase HS, p300, H3K4me1, H3K27ac in *Figure 4—figure supplement 2B and 2C*), which suggested a preferred binding of MOF outside canonical ESC regulatory regions.

In addition to ESC-specific binding of MSL2 and KANSL3 to predicted enhancers, we also identified a very distinct set of TSS-distal binding sites by MSL2 to introns and intergenic regions without enhancer-associated marks (cluster C in *Figure 2*). Approximately, 81% of these cluster C regions had solitary MSL2 enrichments without significant signals of any of the other ChIPed proteins. Interestingly, these MSL2 binding sites increased in number and binding strength upon differentiation to NPCs (829 solitary MSL2 peaks in ESCs compared to 3635 in NPCs). In contrast to the previously described binding sites that were characterized by the prevalence of open, active chromatin (*Figures 3 and 4*), here MSL2 was excluded from hypo-methylated DNA regions (*Figure 4—figure supplement 3A*; note the different behavior of KANSL3). When we searched the unique MSL2 binding sites for DNA motifs, we obtained a $(CAGA)_n$ motif (*Figure 4—figure supplement 3B*) that was previously described as a binding site for SMAD3, a transcription factor that translates the TGF-beta receptor response into gene expression regulation (*Zawel et al., 1998*). When we subsequently scanned all the binding sites for the presence of the published, original SMAD3 motif, we found a strikingly specific signal for the center of the solitary MSL2 ChIP-seq peaks (*Figure 4—figure supplement 3C*).

We conclude that MOF, MSL2 and KANSL3 specifically recognize ESC enhancers. In contrast to MSL-MOF-NSL co-occurrence at housekeeping gene promoters, we found evidence for differential and independent binding of the individual proteins to gene bodies and intergenic regions suggesting the potential for distinct tissue-specific regulatory functions of MSL2 and KANSL3. These data reveal a newly evolved function of MSL2 and KANSL3 in mammals, which has not been observed in flies.

## Genes associated with TSS-distal binding sites of MSL1 and MSL2 are frequently downregulated in cells lacking MSL1 or MSL2

To study the functional implications of the binding of MSL2 and KANSL3 to putative ESC enhancers, we first tested five different regions located near genes related to pluripotency and self-renewal (*Hu et al., 2009*; *Young, 2011*). Using luciferase reporter constructs, we found strong transcriptional enhancement for all tested regions in ESCs, but not in NPCs or 3T3 cells which correlated with the presence of MSL2 and/or KANSL3 and MCRS1 in ESCs only (*Figure 4D*, *Figure 4—figure supplement 4A*).

We then used our RNA-seq data sets from MSL1-, MSL2-, MOF-, and KANSL3-depleted cells to assess the effects on the transcription of those genes that were not bound at promoters, but had been predicted by GREAT (*McLean et al., 2010*) to be regulated by TSS-distal binding sites of the respective protein. As shown in *Figure 4E*, we again found similar effects for KANSL3- and MOF-depleted cells compared to MSL1- and MSL2-depleted cells with the latter group showing genome-wide downregulation of predicted target genes. In fact, the numbers of TSS-distal targets of MSL1 or MSL2 that were significantly reduced in the respective shRNA-treatments were markedly larger than for genes where MSL1 or MSL2 bound to the promoter (compare *Figure 3F* with *Figure 4—figure supplement 4B*). Moreover, in MSL2-, but not KANSL3-depleted cells, the effects on TSS-distally targeted genes were slightly stronger than for TSS-targets (*Figure 4—figure supplement 4C*).

## Depletions of MOF and KANSL3, but not of MSL complex members affect key pluripotency factors

While TSS-binding predominantly occurred at housekeeping genes, we noticed that the majority of enhancer regions associated with key pluripotency factors (e.g., SOX2, ESRRB, MYC, REX1, TBX3, NANOG) were strongly enriched for MSL2 and KANSL3. We thus assessed the effects of the protein depletions on pluripotency factors in ESCs and found strongly reduced levels of NANOG, REX1, and ESRRB in MOF- or KANSL3-depleted cells. Surprisingly, the pluripotency factors remained almost unaffected in cells depleted of MSL1 or MSL2 (*Figure 4F*). These contrasting results were mirrored by decreased levels of alkaline phosphatase (AP) in MOF- and KANSL3-, but not in MSL1- or MSL2-depleted cells (*Figure 4—figure supplement 4D*).

These findings indicate that despite their frequent effects on TSS-distally targeted genes, MSL1 and MSL2 might not show dominant effects at genes that are bound by KANSL3 as well. Therefore, we specifically searched for regions without KANSL3 binding to identify putative MSL-specific functions.

## The MSL complex binds multiple loci within the X inactivation center

As described previously, we identified only a small subset of regions in the mouse genome where MSL complex members were enriched exclusively (see cluster E in *Figure 2*). Strikingly, several of these binding sites fall into a region known as the X inactivation center (XIC). The XIC is the X-chromosomal region necessary and sufficient to control the inactivation of one of the two X chromosomes in females (reviewed in *Pollex and Heard, 2012*).

The XIC site with the strongest concomitant enrichments of MSL1, MSL2 and MOF was the major promoter (P2) of *Tsix* and its intronic minisatellite—*DXPas34* (*Figure 5A,B*). *DXPas34* is a well-characterized tandem repeat that serves as a binding platform for multiple transcription factors and contains bidirectional enhancing properties essential for the expression of *Tsix*, the antisense transcript of *Xist* (*Debrand et al., 1999*; *Cohen et al., 2007*; *Donohoe et al., 2007*; *Navarro et al., 2010*; *Gontan et al., 2012*). In rodents, *Tsix* antisense transcription across the *Xist* promoter is required for regulating the levels of *Xist* accumulation. In turn, *DXPas34* deletion impairs the recruitment of Pol II and TFIIB to the major promoter of *Tsix* causing its downregulation (*Vigneau et al., 2006*).

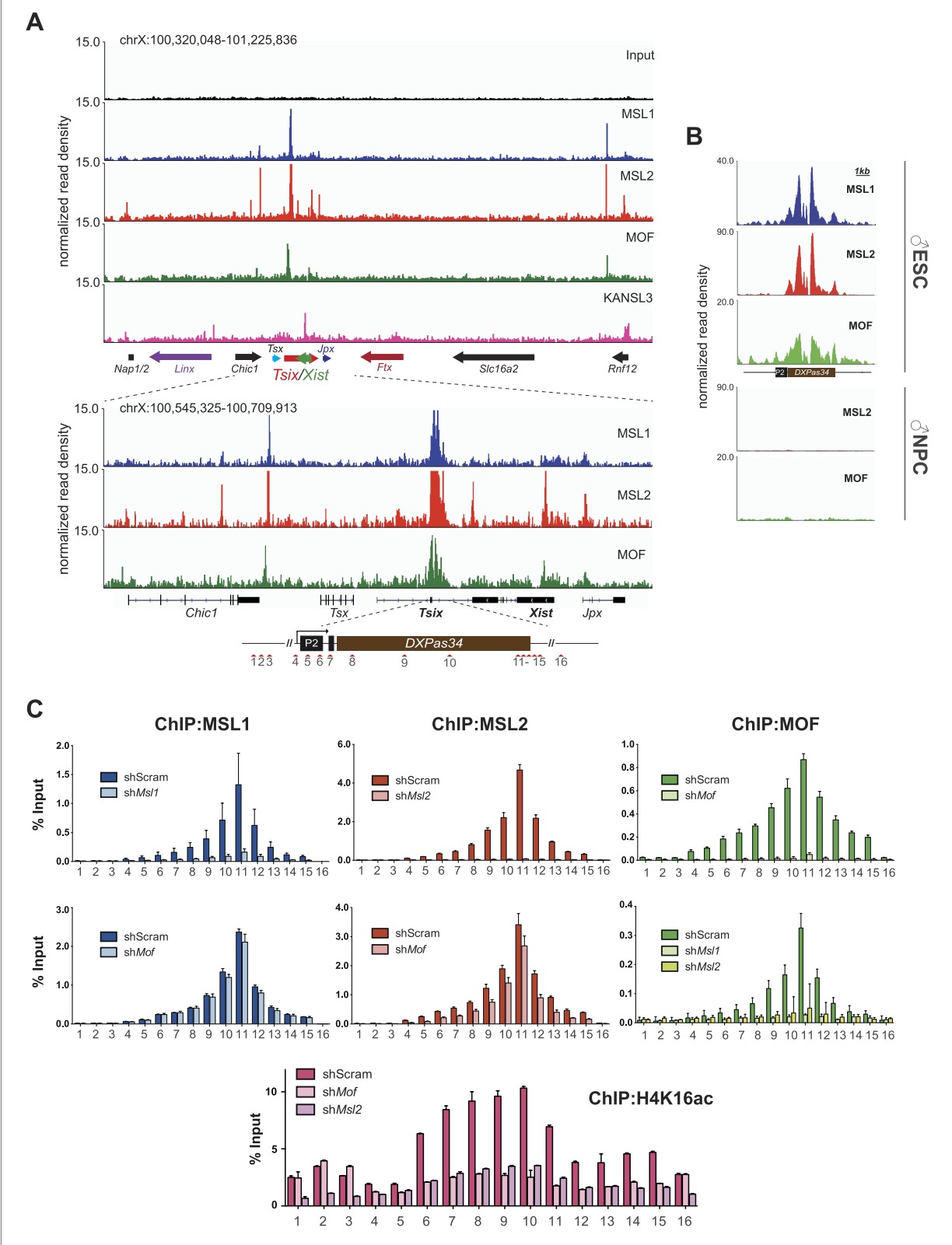

**Figure 5**. The MSL complex binds multiple loci within the X inactivation center including the *Tsix DXPas34* minisatellite enhancer. (**A**) Genome browser snapshots of the mouse X inactivation center (approximately 0.9 Mb) (upper panel) plus enlargement of the 164 kb region between *Chic1* and *Jpx/Enox* (lower panel). The signals shown are the sequencing-depth normalized profiles for ChIP-seq from ESCs (for corresponding profiles in NPCs see *Figure 5. Continued on next page*

*Figure 5. Continued*

*Figure 5—figure supplement 1A*); colored arrows indicate genes of lncRNAs. The schematic representation of the *DXPas34* locus depicts the locations of the primer pairs that were used for ChIP-qPCR analyses (*Supplementary file 3B*). (**B**) Genome browser snapshots of the *DXPas34* minisatellite of sequencing-depth normalized ChIP-seq profiles in ESCs and NPCs. (**C**) ChIP-qPCR analyses of MSL1 (blue), MSL2 (red), MOF (green), and H4K16 acetylation (purple) across the *Tsix* major promoter (*P2*) and the *DXPas34* enhancer in male ESCs treated with the indicated shRNAs. For corresponding ChIP-qPCR in female ESCs see *Figure 5—figure supplement 1C*. Panels in the middle show the effects of MOF depletion on the recruitment of MSL1 and MSL2 to *DXPas34* and vice versa. The bottom panel shows effects of depletion of control (dark pink), MOF (light pink) and MSL2 (purple) on the H4K16 acetylation signal. The labels of the x axes correspond to the arrowheads in (**A**). Results are expressed as mean ± SD of three biological replicates; cells were harvested on day 4 (*Msl1*, *Msl2*) or 5 (*Mof*) after shRNA treatment. For primer pairs see *Supplementary file 3C*.

The following figure supplements are available for figure 5:

**Figure supplement 1**. The MSL proteins bind to multiple loci within the X inactivation center (XIC).

In addition to the *DXPas34* binding site, we detected MSL peaks on the promoters, gene bodies and intronic regions of other key XIC regulators including the genes of the long non-coding (lnc) RNAs *Xist* and *Jpx*. Additionally, we observed peaks upstream of the *Tsx* gene and both at the TSS and downstream of the *Rnf12* gene (*Figure 5A*). Products of all of these genes were shown to play important roles in orchestrating the process of X inactivation (*Stavropoulos et al., 2001*; *Shin et al., 2010*; *Tian et al., 2010*; *Anguera et al., 2011*; *Chureau et al., 2011*; *Gontan et al., 2012*; *Sun et al., 2013*).

The XIC binding of MSL-MOF was specific to ESCs, as almost all enrichments were abolished upon differentiation, except for some loci upstream of *Xist* where traces of binding could still be detected in NPCs (e.g., *Ftx* and *Jpx* TSS, *Figure 5—figure supplement 1A*).

We next confirmed the high ChIP-seq enrichments of MSL1, MSL2 and MOF and assessed H4K16 acetylation on the major promoter of *Tsix* and along *DXPas34* with ChIP-qPCR assays covering the entire region in male and female ESCs (*Figure 5C*, *Figure 5—figure supplement 1B*). Interestingly, the recruitment of MOF was almost completely abolished in both MSL1- and MSL2-depleted cells, whereas the depletion of MOF had no effect on MSL1 and MSL2 binding to the *Tsix* major promoter and *DXPas34* (*Figure 5C*). H4K16 acetylation ChIP signals were severely reduced in both MOF- and MSL2-depleted cells. These results are in agreement with our global observations (*Figure 3G*, *Figure 3—figure supplement 4C*) and indicate that MSL1 and MSL2 are together necessary and sufficient for the recruitment of MOF and for the deposition of H4K16 acetylation at *DXPas34*.

## MSL1 and MSL2 are important for *Tsix* expression

To directly assess the functional outcome of MOF-, MSL1-, and MSL2-depletions, we studied the expression of *Tsix* and *Xist* in shRNA-treated ESCs. Unexpectedly, only MSL1- and MSL2-, but not MOF-depletion led to pronounced downregulation of *Tsix* both in male and female ESCs (*Figure 6A*; note that in our RNA-seq data set for MSL2-depleted cells, *Tsix* was among the five most strongly downregulated genes). Downregulation of *Tsix* was accompanied by moderately elevated *Xist* RNA levels in MSL1- and MSL2-depleted ESCs whereas depletion of MOF yielded the most pronounced (8–15-fold) upregulation of *Xist* without affecting *Tsix*.

To determine the effects on *Tsix* in individual cells, we next performed RNA-FISH with probes against *DXPas34* and *Huwe1* in female ESCs (*Huwe1* was used to mark X chromosomes, for probe references see 'Materials and methods'). The RNA-FISH confirmed the qPCR results as we observed global reduction and in many cases elimination of *DXPas34* signals in MSL1- and MSL2-, but not in MOF-depleted cells (*Figure 6B*, *Figure 6—figure supplement 1A–C*).

We next wanted to understand the mechanistic differences between the *Tsix*-specific and the *Tsix*-independent effects on *Xist* levels that we found for depletions of MSL1/MSL2 and MOF, respectively. As pluripotency factors are additional regulators of *Xist* (*Navarro et al., 2008*; *Nesterova et al., 2011*), we assessed the consequences of the different knockdowns on the *Xist*-related pluripotency network in female ESCs. Like for MOF- and KANSL3-depletions in male ESCs (*Figure 4F*), the depletion of MOF (but not of MSL1 or MSL2) in female ESCs resulted in a significant decrease of transcript and protein levels of pluripotency factors that had previously been associated with *Xist* repression (e.g., NANOG and REX1; see *Figure 6C*, *Figure 6—figure supplement 1D*).

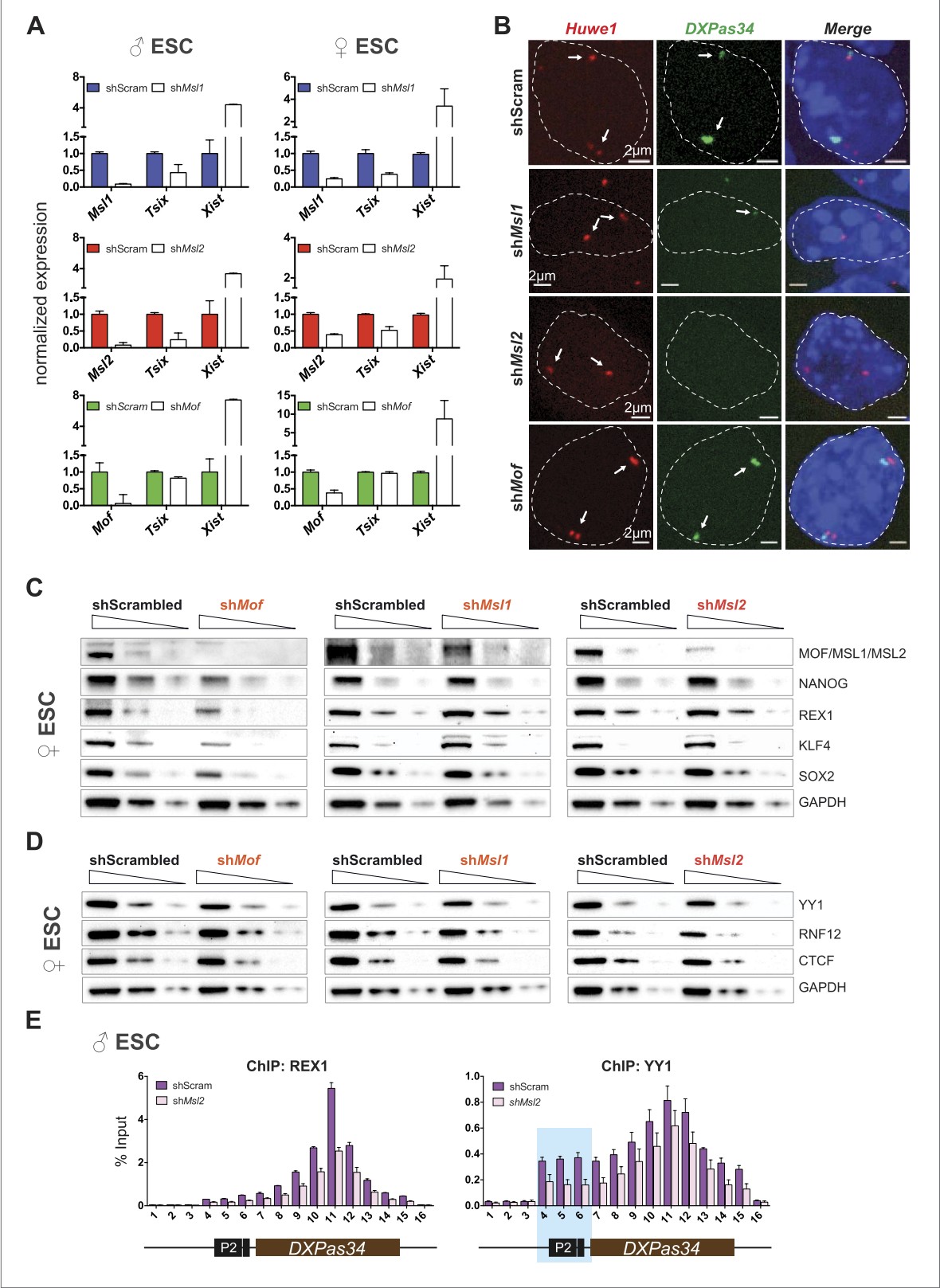

**Figure 6**. Depletion of MSL1 and MSL2 leads to downregulation of *Tsix* with concomitant upregulation of *Xist*. (**A**) Gene expression analysis for the indicated genes in male and female ESCs treated with scrambled RNA (shScram) or shRNA against *Msl1*, *Msl2*, or *Mof*. All results are represented as relative values normalized to expression levels in shScram (normalized to *Hprt*) and expressed as means ± SD in three biological replicates. (**B**) RNA-FISH

*Figure 6. Continued on next page*

*Figure 6. Continued*

for *Huwe1* (red) and *DXPas34* (green) in: scrambled control, sh*Msl1*-, sh*Msl2*-, and sh*Mof*-treated female ESCs. Nuclei were counterstained with DAPI (blue). White arrows denote foci corresponding to *Huwe1* or *Tsix*; dashed lines indicate nuclei borders. For additional images, phenotypes and quantifications see *Figure 6—figure supplement 1A–C*. For probe references see 'Materials and methods'. (**C**) Western blot analyses of the pluripotency factors in scrambled-, *Mof*-, *Msl1*-, and *Msl2*-shRNA-treated female ESCs. For corresponding expression analyses see *Figure 6—figure supplement 1D,E*. The respective dilution (100%, 30%, 10%) of loaded RIPA extracts is shown above each panel. GAPDH was used as the loading control. For antibodies see 'Materials and methods'. (**D**) Western blot analyses of the transcription factors involved in regulation of the XIC in scrambled-, *Mof*-, *Msl1*-, and *Msl2*-shRNA-treated female ESCs. The respective dilution (100%, 30%, 10%) of loaded RIPA extracts is shown above each panel. GAPDH was used as the loading control. (**E**) ChIP-qPCR analysis of REX1 (left panel) and YY1 (right panel) across the *Tsix* major promoter (P2) and *DXPas34* in male ESCs treated with the indicated shRNAs. The labels of the x axes correspond to the arrowheads in *Figure 5A*. For all ChIP experiments, three biological replicates were used; results are expressed as mean ± SD; cells were harvested on day 4 (*Msl2*) or 5 (*Mof*) after shRNA treatment.

The following figure supplements are available for figure 6:

**Figure supplement 1**. Cells depleted of MSL1 or MSL2, but not MOF show loss of *DXPas34* foci.

Taken together, we detect direct binding of MSL complex members to several loci within the X inactivation center including the *Tsix/Xist* locus. Depletion of MSL1 or MSL2, but not MOF led to severe downregulation of *Tsix* expression while depletion of MOF, MSL1, or MSL2 resulted in elevated *Xist* levels. These results indicate a direct regulatory function of MSL1 and MSL2 on the *DXPas34* locus and an indirect NSL-associated MOF effect on *Xist* expression through the pluripotency network.

## Depletion of MSL1 and MSL2 leads to impaired recruitment of REX1 and YY1 to regulatory regions of *Tsix*

As loss of MSL1 and MSL2 did not affect the core pluripotency network, we set out to explore what might be the impact of MSL depletion on XIC genes (other than *Tsix* and *Xist*) and transcription factors involved in their regulation. As shown in *Figure 6—figure supplement 1E*, we observed mild effects on the expression of XIC-encoded genes involved in the regulation of X inactivation. Only depletion of MSL2 led to significant downregulation of *Ftx* and *Jpx* genes whose promoters were bound by MSL1 and/or MSL2 (*Figure 5A*). On the other hand, depletion of MOF led to moderate upregulation of *Linx* lncRNA, which acts synergistically with *Tsix* (*Nora et al., 2012*).

Neither the depletion of MSL1 and MSL2 nor the depletion of MOF significantly influenced protein levels of RNF12, YY1, or CTCF that are known regulators of the XIC (*Figure 6D*; *Donohoe et al., 2007*, *2009*; *Jonkers et al., 2009*; *Shin et al., 2010*; *Jeon and Lee, 2011*). Since REX1 and YY1 bind and regulate the *Tsix* locus (*Donohoe et al., 2007*; *Gontan et al., 2012*), we subsequently tested whether MSL depletion would affect the recruitment of these factors to the *Tsix* major promoter and *DXPas34*. Indeed, the depletion of MSL2 led to significant reduction of REX1 ChIP signals across the *DXPas34* locus whereas the effect on YY1-targeting was less pronounced and restricted to the *Tsix* major promoter (P2) (*Figure 6E*).

## Knockdown of *Msl1* and *Msl2* results in enhanced accumulation of *Xist* and X-chromosomal coating in differentiating female ESCs

We next assessed the consequence of MSL-dependent reduction of *Tsix* levels and concomitant upregulation of *Xist* at a cellular level using RNA-FISH for *Xist* upon depletion of individual MSL complex members (for probe reference see 'Materials and methods'). Interestingly, we observed accumulating *Xist* lncRNA and X-chromosomal coating in a small fraction of MSL1- and MSL2-depleted female ESCs (but not MOF-depleted cells; 4–5% of the cell population in sh*Msl1* and sh*Msl2* with comparison to 0.5% in scrambled control, see *Figure 7A* and *Figure 7—figure supplement 1A–C*). These findings suggest that the MSL1- and MSL2-dependent downregulation of *Tsix* is sufficient to cause occasional accumulation of *Xist* lncRNA in undifferentiated female ESCs. The different outcomes following MOF and MSL1/MSL2 depletion on *Xist* confirmed the notion that MOF and MSL1/MSL2 influence the XIC via different mechanisms.

Previous studies have shown that the effects of *Tsix* depletion on *Xist* accumulation and X inactivation become fully apparent after induction of differentiation (*Clerc and Avner, 1998*; *Debrand et al., 1999*; *Lee and Lu, 1999*; *Luikenhuis et al., 2001*; *Ohhata et al., 2006*; *Sun et al., 2006*).

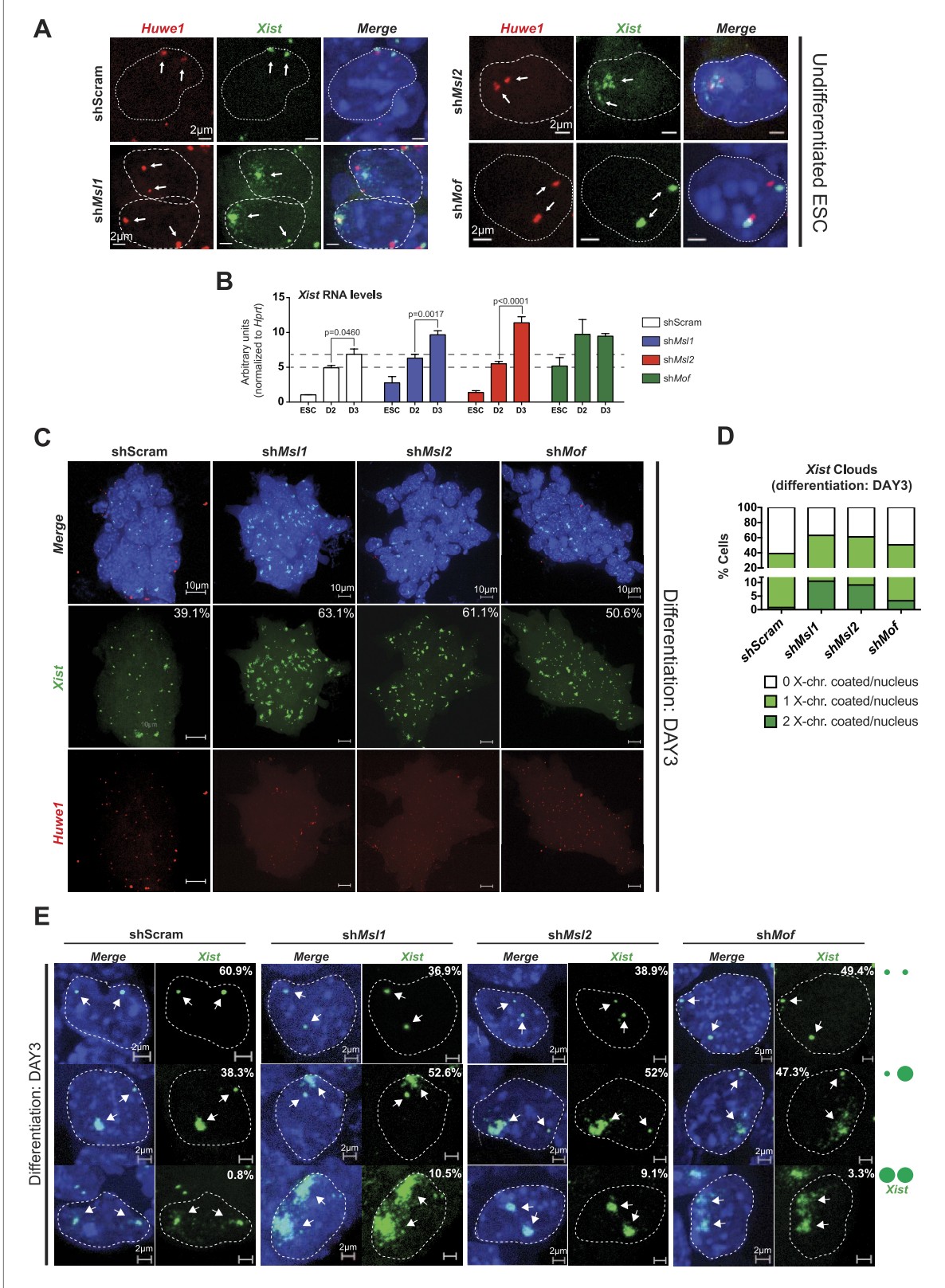

**Figure 7**. MSL1 and MSL2 depletion leads to enhanced and chaotic *Xist* accumulation in early differentiation. (**A**) RNA-FISH for *Huwe1* (red) and *Xist* (green) in: scrambled control, sh*Msl1*-, sh*Msl2*-, and sh*Mof*-treated female ESCs. Nuclei were counterstained with DAPI (blue). White arrows denote foci corresponding to *Huwe1* or *Xist*; dashed lines indicate nuclei borders. For additional images, phenotypes and quantifications see *Figure 7—figure supplement 1B–D*.
*Figure 7. Continued on next page*

*Figure 7. Continued*

For probe references see 'Materials and methods'. (**B**) Expression analysis for *Xist* in undifferentiated, day 2 (D2) and day 3 (D3) differentiating female ESCs treated with scrambled RNA (shScram) or shRNA against *Mof, Msl1,* and *Msl2*. All results are represented as arbitrary units (*Xist* expression in undifferentiated ESCs = 1) normalized to expression levels in shScram (normalized to *Hprt*) and expressed as means ± SD in three biological replicates. p-values for D2-to-D3 expression change were obtained using unpaired *t* test. (**C**) RNA-FISH for *Huwe1* (red) and *Xist* (green) in: scrambled control, sh*Msl1-*, sh*Msl2-*, and sh*Mof*-treated differentiating female ESCs. Nuclei were counterstained with DAPI (blue). RNA-FISH was performed on the sixth day of knockdown (after 72 hr of differentiation). Percentages indicate number of cells with at least one *Xist* cloud for each of the knockdowns. For additional images of multicellular colonies see *Figure 7—figure supplement 2A*. (**D**) Bar plot summarizing the percentage of *Xist* clouds for individual knockdowns in differentiating (DAY3) female ESCs for individual knockdowns. Cells were divided into three categories: cells carrying no *Xist* clouds (white), single *Xist* cloud (light green), or two *Xist* clouds (dark green). For quantifications, see *Figure 7—figure supplement 2B*. (**E**) RNA-FISH for *Xist* (green) in: scrambled control, sh*Msl1-*, sh*Msl2-*, and sh*Mof*-treated differentiating (DAY3) female ESCs. Here, we show examples of individual nuclei carrying different patterns of *Xist* accumulation. Percentages correspond to the frequency of the shown *Xist* pattern within the population of cells. White arrows denote *Xist* foci; dashed lines indicate nuclei borders. For quantifications see *Figure 7—figure supplement 2B*.

The following figure supplements are available for figure 7:

**Figure supplement 1**. Depletion of MSL1 and MSL2 leads to occasional accumulation and spreading of *Xist* in undifferentiated ESCs.

**Figure supplement 2**. Depletion of MSL1 and MSL2 lead to enhanced *Xist* accumulation in differentiating ESCs.

We therefore depleted MSL1, MSL2 and MOF, and induced differentiation for 3 days by withdrawing LIF and placing the ESCs in N2B27 media. Consistent with our previous results, the induction of differentiation resulted in a stronger elevation of *Xist* RNA levels in MSL1- and MSL2-depleted cells in comparison to the scrambled control (*Figure 7B*). As *Tsix* expression was not affected in MOF-depleted ESCs and *Xist* levels were already high before induction of differentiation, *Xist* upregulation between day 2 and 3 of differentiation was similar to the scrambled control.

To monitor the effect on the X chromosome more closely, we next performed *Xist* RNA-FISH in MSL1-, MSL2- and MOF-depleted cells after 3 days of differentiation. All three knockdowns resulted in enhanced *Xist* accumulation and X-chromosomal coating (63.1%, 61.1% and 50.6% of all counted cells in sh*Msl1-,* sh*Msl2-,* and sh*Mof*-treated ESCs, respectively, in comparison to scrambled control with 39.1% of counted cells; see *Figure 7C,D* and *Figure 7—figure supplement 2A,B*). Interestingly, we observed that MSL1- and MSL2-depleted differentiating cells contained numerous cells with two inactive X chromosomes. The fraction of cells where both X chromosomes underwent XCI was approximately 10-fold higher in *Msl1* and *Msl2* knockdown compared to the scrambled control (*Figure 7E*). These results are in agreement with previously published data from homozygous *Tsix* mutants that exhibit irregular, 'chaotic' choice for X inactivation (*Lee, 2005*).

Taken together, our data establishes MSL1 and MSL2 among the key regulators of *Tsix* transcription, as the depletion of MSL proteins results in severe downregulation of *Tsix* transcription and enhanced accumulation of *Xist* during early differentiation.

## Discussion

We present a thorough characterization of the histone acetyltransferase MOF and its two known complexes in mouse embryonic stem cells (ESCs) and neuronal progenitor cells (NPCs). We determined five basic modes of co-occurrence that revealed cell-type-specific as well as constitutive functions of the different proteins and support the notion that the NSL complex has general, housekeeping functions whereas the MSL complex predominantly performs more specialized tasks. We show that MOF and its associated proteins are involved in gene expression regulation via different means: first, they all target the promoters of housekeeping genes in a cell-type-independent manner and second, members of both complexes occupy different sets of ESC-specific enhancers that are essential for the maintenance of stem cell identity. We demonstrate the distinct and novel functions carried out by the MOF-associated complex members by revealing that both complexes contribute to the repression of X inactivation in ESCs via different means: While we establish the MSL complex as a direct regulator of *Tsix*, MOF and the NSL complex play an important role in the maintenance of pluripotency factors (*Figure 8*).

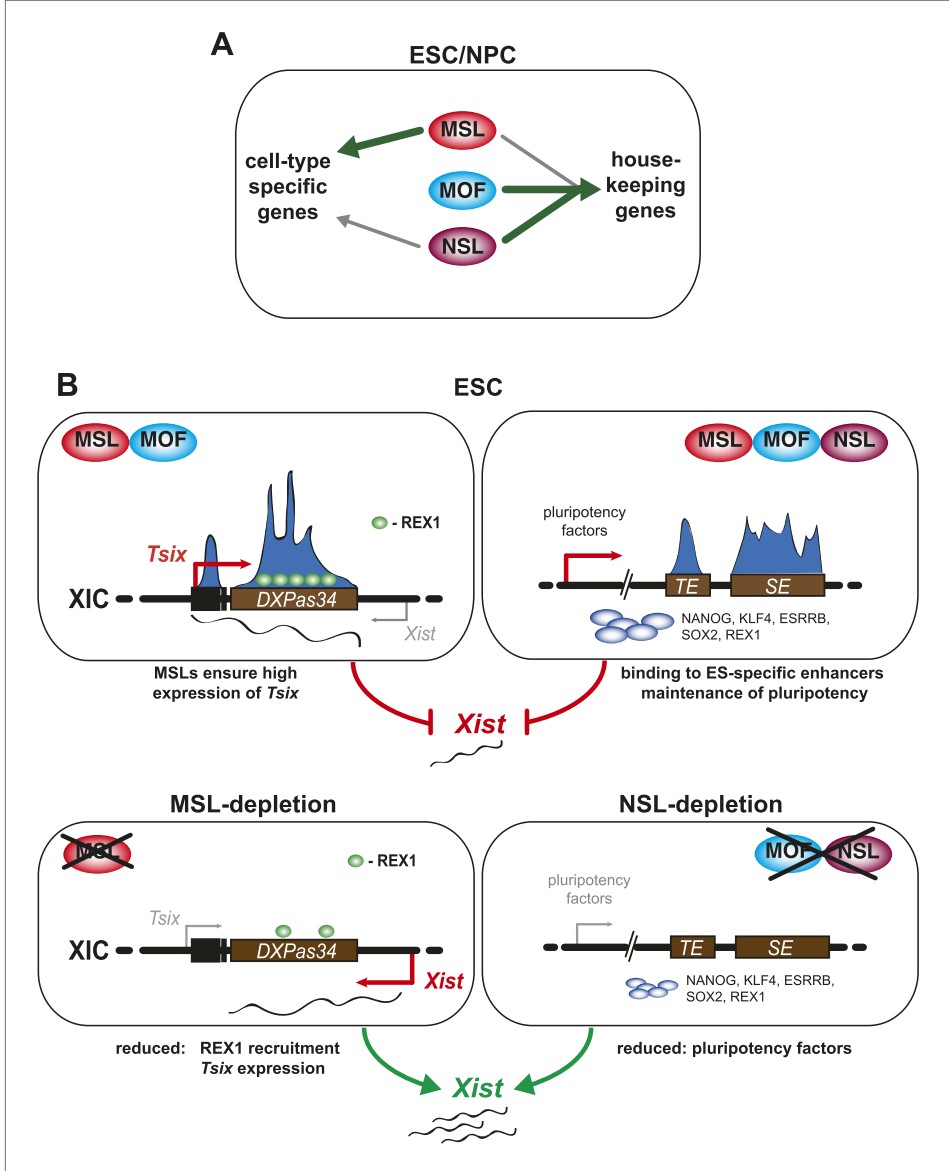

**Figure 8**. A summary model. Shared and distinct pathways by which MOF, MSLs and NSLs regulate gene expression, pluripotency, and the X inactivation center. (**A**) In this study, we have identified several modes of concurrent and independent binding of mammalian MOF, MSL and NSL proteins. We find that all complexes bind to promoters of housekeeping genes in ESCs and NPCs with NSL complex members occupying the majority of the target genes, while MOF and MSL proteins bind NSL-bound genes in a more restricted manner. Furthermore, we observe that upon differentiation, KANSL3 and MSL2 additionally occupy TSSs of different sets of cell-type-specific genes in the absence of MOF. (**B**) When we studied the functions of MSL and NSL complexes at the murine X inactivation center, we determined two basic mechanisms by which the different proteins affect the maintenance of two active X chromosomes in ESCs. (1) MSLs bind to the promoter and enhancer of *Tsix* whose transcription represses *Xist* expression. Upon depletion of MSLs, *Tsix* expression is compromised, so is REX1 recruitment to the *Tsix* locus. Consequently, *Xist* is increasingly transcribed and can occasionally accumulate. (2) In addition, MOF, MSLs, and NSLs bind to typical enhancers (TE) and super enhancers (SE) in ESCs, and notably those of pluripotency factors. In WT ESCs, the high expression of pluripotency factors is another layer of *Xist* repression. The depletions of MOF or KANSL3, but not of MSL1 or MSL2 reduce the expression of pluripotency factors involved in *Xist* repression causing a *Tsix*-independent increase of *Xist* expression.

## Global effects of MOF are correlated with the NSL complex

Our study sheds light on the interplay between MOF and its complexes in mammals. Despite the fact that the depletion of KANSL3 does not strongly reduce global H4K16 acetylation levels, we

observed strikingly similar protein and transcriptome changes in KANSL3- or MOF-depleted cells (*Figure 3E–G*). On the other hand, MSL1- and MSL2-depletion caused marked decreases of H4K16 acetylation (*Figure 3G*). This is consistent with previous reports that established MSL proteins as the main enhancers of MOF's H4K16 acetylation activity, while the NSL complex was shown to possess broader substrate specificity and can crosstalk with histone methylases (*Cai et al., 2010*; *Kadlec et al., 2011*; *Zhao et al., 2013b*). Unexpectedly, we observed remarkably different phenotypic changes in MSL1- or MSL2-depleted cells compared to MOF- and KANSL3-depleted cells (*Figure 3E*, *Figure 3— figure supplement 3A*). A striking example was the strong reduction of key pluripotency factors in KANSL3- and MOF-depleted cells that remain unaffected in MSL1- and MSL2-knockdowns (*Figure 3G*, *Figure 4F*). These results support the recent finding that MOF is vital for the maintenance of pluripotency (*Li et al., 2012*), but we furthermore show that this is an NSL- and not MSL-related function of MOF independent of H4K16 acetylation deposition.

Taken together, our data shows that while MOF is the major acetyltransferase for lysine 16 of histone 4 (*Taipale et al., 2005*), MSL-dependent H4K16 acetylation is one of several means through which MOF exerts its crucial biological functions. This notion was further supported by the finding that MOF predominantly binds to promoters of broadly expressed genes as part of the NSL complex and subsequently supports their transcription (*Figure 3A–F*). MSL1 and MSL2, on the other hand, bound to a relatively small subset of broadly expressed MOF-NSL-targeted genes that were significantly stronger expressed than those where MOF was exclusively present with NSL complex members (*Figure 3B*, *Figure 3—figure supplement 1B*). The additive effects of the complexes on gene expression were intriguing, and whether they influence each other's activity or exert their functions separately should be studied in the future. We propose that the MSL complex fine-tunes MOF's activity and ensures precise regulation of more specific targets—after all, their presence is essential for the recruitment of MOF to NSL-independent targets (*Figure 5B*). Our model is surprisingly similar to the picture that is emerging from *Drosophila* research where the NSL complex regulates housekeeping genes (*Feller et al., 2012*; *Lam et al., 2012*) while the MSL complex fulfills a highly specialized role on the male X chromosome (reviewed in *Conrad and Akhtar, 2011*).

## MSL2 and KANSL3 can contribute to transcription via enhancer binding

In addition to insights about MOF-related functions of MSL and NSL complexes, we show for the first time additional binding of MSL and NSL proteins to TSS-distal regions with enhancer characteristics. On a global scale, MOF did not yield strong enrichments for canonical enhancers; however, both MSL2 and KANSL3 showed robust signals for TSS-distal regions in ESCs, but not in NPCs, which reflected the transcriptional activity of these regions (*Figure 2*, *Figure 4A*). This apparent MOF-independent binding of the individual proteins (that tended to prefer different sets of enhancers; *Figure 4B*) suggests that KANSL3 and MSL2 stimulate transcription even in the absence of the histone acetyltransferase. Both proteins are in principle capable of supporting transcription: the *Drosophila* homologue of KANSL3 can directly activate transcription in vitro (*Raja et al., 2010*) and human MSL2 acts as an E3 ubiquitin ligase at lysine 34 of H2B (H2BK34ub) (*Wu et al., 2011*), which has been suggested to promote methylation of H3K4, and thus gene expression (*Wu et al., 2011*). Indeed, we observed several hundred genes that had been predicted to be regulated by TSS-distal binding sites of MSL2 or KANSL3 to be downregulated in the respective knockdowns with particularly high frequencies in MSL2-depleted cells (*Figure 4E*). It is important to note that the subset of ESC enhancers for key pluripotency factors (e.g., *Klf4*, *Sox2*) were bound concomitantly by KANSL3 and MSL2 and only the depletion of KANSL3, but not of MSL1 or MSL2 diminished protein and transcript levels of these key ESC molecules (see above). It is possible that KANSL3 could rescue loss of MSL2 at certain loci, but the exact mechanisms through which KANSL3 affects transcription via enhancer-binding need to be studied further. Furthermore, the pluripotency network and/or Mediator-related functions at super enhancers may be sufficient and dominant over MSL2 to maintain the expression of the pluripotency factors in the absence of MSL2, but may well be dependent on the function of KANSL3 at these regions.

## MSL1 and MSL2 repress X inactivation by regulating *Tsix* expression

When we specifically searched for regions where KANSL3 was not present together with MSL1 and MSL2, we found that the X inactivation center (XIC) showed numerous signals for the MSL complex (*Figure 5*). The XIC, a hot-spot of regulatory lncRNAs, is an X-chromosomal region that contains the main regulators of X chromosome inactivation (XCI). The proper function of XIC-located non-coding

RNAs is influenced by the spatial organization of the XIC and governed by a sophisticated interplay of multiple transcription factors such as pluripotency factors (*Donohoe et al., 2007*; *Navarro et al., 2010*; *Deuve and Avner, 2011*; *Gontan et al., 2012*; *Nora et al., 2012*).

We found that depletion of MSL1 and MSL2 severely reduced *Tsix* expression in male and in female ESCs, moderately increased *Xist* levels (*Figure 6A*), but left pluripotency factors unaffected (*Figure 6C*). In contrast, MOF-depleted cells showed downregulation of pluripotency factors and much higher *Xist* levels. Previous studies demonstrated that in undifferentiated ESCs, where pluripotency factors are highly abundant, even severe downregulation of *Tsix*, or *Tsix*-deletion has almost no effect on *Xist* transcription (*Morey et al., 2001*; *Navarro et al., 2005*; *Nesterova et al., 2011*). Thus, the pronounced *Xist* upregulation seen in MOF-depleted cells seems to be an indirect effect due to the downregulation of pluripotency factors, while the reduction of *Tsix* transcripts in MSL1- and MSL2-depleted cells, where pluripotency factors remain unaffected, has milder consequences on *Xist* levels.

Consequently, we could show that once ESCs are forced to initiate differentiation, the depletion of MOF has mild effects while MSL1- and MSL2-depleted cells, in which *Tsix* expression is prematurely downregulated, indeed suffer from enhanced *Xist* accumulation accompanied by 'chaotic' X inactivation (different numbers of inactivated X chromosomes within a population of cells; *Figure 7B–E*). This is consistent with the notion that the repressive potential of *Tsix* on *Xist* accumulation and the role of *Tsix* and the *DXPas34* locus in the process of counting and choice of XCI (*Lee, 2005*; *Vigneau et al., 2006*) becomes fully apparent during early stages of differentiation where additional repressive factors such as pluripotency factors are downregulated (reviewed in *Rougeulle and Avner, 2004*).

## Conclusion

We show that NSL and MSL complex members can function in concert to ensure proper regulation of gene expression, but our findings also strongly imply that members of both complexes have the capacity to act independently. In the case of the X inactivation center, we observe that the MOF-interacting proteins, despite engaging different regulatory means (MSL1, MSL2 through direct regulation of *Tsix*, and MOF-NSL through the pluripotency network) synergize to ensure the proper expression of the X chromosomes in undifferentiated ES cells (*Figure 8*). Our study sets the ground for future research to dissect the intricate interactions and specific functions of MOF and its associated major regulatory proteins in more detail.

## Materials and methods

### Cell culture

All cell culture was performed in a humidified incubator at 37°C and 5% $CO_2$. The feeder-dependent mouse female embryonic stem cell line F1-21.6 was cultivated on mitomycin-C-inactivated or irradiated mouse embryonic fibroblasts (MEFs). The feeder-independent mouse male ES cell line WT26, a kind gift from the lab of Thomas Jenuwein, was cultivated on gelatin-coated dishes in ESC culture media KnockOut-DMEM (Gibco, Carlsbad, CA) supplemented with 1% L-glutamine (Gibco), 1% penicillin/streptomycin (Gibco), 1% non-essential amino acids (Gibco), 1% sodium pyruvate, 1% 2-mercaptoethanol. All ESC media contained 15% FBS and 1000 U/ml (for feeder-dependent) or 2000 U/ml (for feeder-) of leukemia inhibitory factor.

Male and female neuronal progenitor cell (NPC) lines were derived from previously mentioned ES cell lines (see below). Mouse 3T3 cells (for luciferase assays) and human HEK293-FT cells (for lentiviral production) were cultivated in DMEM (high glucose, with glutamine, Gibco) supplemented with 10% heat-inactivated serum (PAA Laboratories, North Dartmouth, MA), 1% L-glutamine, 1% penicillin/streptomycin.

### NPC differentiation

Mouse ESCs were differentiated into neuronal progenitor cells (NPC) as previously described (*Conti et al., 2005*; *Splinter et al., 2011*). In brief, $1 \times 10^6$ ESCs (deprived of feeder cells) were plated on 0.1% gelatin-coated dishes in N2B27 medium and cultured for 7 days with daily media changes. The cells were then dissociated from the plate using accutase (Sigma, Germany) and $3 \times 10^6$ cells were plated on a bacterial petri dish to induce formation of embryoid bodies in N2B27 medium supplemented with 10 ng/ml EGF and FGF2 (Peprotech, Rocky Hill, NJ). After 72 hr, embryoid bodies were transferred to 0.1% gelatin-coated dishes to allow adhesion and expansion of NPCs from the embryoid bodies. NPC lines were maintained in N2B27 medium supplemented with EGF and FGF2

(10 ng/ml each), on 0.1% gelatin-coated flasks. For FISH analysis, F1-21.6 ESCs were grown on gelatin-coated coverslips with a MEF-inactivated monolayer for 24 hr.

## Western blot analysis

The Invitrogen precast gel system NuPAGE was used for SDS-PAGE. The 4–12% Bi–Tris gradient gels (for proteins above 20 kDa) or 12% Bis–Tris gels (for histones and histone marks) were loaded with samples supplemented with Roti-Load 1 sample buffer. After blotting, the membranes were blocked in 5% milk with PBS + 0.3% Tween-20 (PBST) mix for at least 1 hr at room temperature. Membranes were then incubated overnight with the primary antibody in 0.5% milk with PBST at 4°C. The next day, membranes were washed three times for 10 min in PBST, incubated with a suitable HRP-coupled secondary antibody for 1 hr at room temperature, washed thrice and proteins were visualized with Lumi-Light Plus Western Blotting Substrate using the Gel Doc XR+ System.

## Immunoprecipitation assays (IP and ChIP)

For (co)immunoprecipitation (IP, co-IP) experiments, 1 ml of nuclear extract (0.5 mg/ml) was used. IPs were performed in IP buffer (25 mM HEPES pH 7.6, 150 mM KCl, 5 mM MgCl2, 0.5% Tween20, 0.2 mg/ml BSA, 1× complete protease inhibitors tablet). Extracts were incubated with 5 µg of the respective antibody or normal-rabbit/normal rat serum. For MSL1 15 µl of antibody serum was used. Extracts were incubated with the antibody for 2 hr, rotating at 4°C. Protein-A Sepharose beads (GE Healthcare, United Kingdom), blocked with 1 mg/ml yeast tRNA and 1 mg/ ml BSA (NEB, Ipswich, MA), were used for all ChIP and IP assays.

Chromatin immunoprecipitation (ChIP) assays were performed as previously described (*Pauli, 2010*) with minor changes. Cells were fixed in 1% molecular biology grade formaldehyde (Sigma) 9 min before being quenched with glycine (0.125 M final concentration). Cells were washed twice with ice-cold PBS and lysed on ice for 10 min with 10 ml of Farnham lysis buffer (5 mM PIPES pH 8.0, 85 mM KCl, 0.5% NP-40 + Roche Protease Inhibitor Cocktail Tablet, filtered through 0.2 micron filter unit). Lysates were transferred to a Kontes dounce tissue grinder (K885300-0015, size B) and dounced 15 times in order to break the cells and keep nuclei mostly intact. Crude nuclear prep was transferred to 15-ml falcon tube and nuclei pelleted by centrifugation at 2000 rpm at 4°C for 5 min. Nuclei were resuspended in RIPA lysis buffer (1 × PBS, 1% NP-40, 0.5% sodium deoxycholate, 0.1% SDS + Roche Protease Inhibitor Cocktail Tablet, filtered through 0.2 micron filter unit). The nuclear extract was subjected to chromatin shearing using the Diagenode Bioruptor Plus sonicator (at high setting for a total time of 25 min, 30 s ON, 30 s OFF). The sonicated mixture was centrifuged at 14,000 rpm at 4°C for 5 min and supernatant was collected. Chromatin was supplemented with 5 µg of primary antibody and incubated for 16 hr (antibodies used for ChIP are listed below). After incubation, 50 µl of 50% slurry bead solution was added for another incubation period (2 hr), then beads were washed: four times for 15 min with RIPA lysis buffer, two times for 1 min with LiCl IP wash buffer (250 mM LiCl, 10 mM Tris–HCl pH 8.0, 1 mM EDTA, 0.5% NP-40, 0.5% DOC, filtered through 0.2 micron filter unit), two times for 1 min with TE buffer (1 mM Tris–HCl pH 8.0, 1 mM EDTA, filtered through 0.2 micron filter unit). Washed beads were resuspended in 100 µl of IP elution buffer and subjected to overnight reverse cross-linking (RNase and proteinase K digestions) followed by DNA purification (DNA was purified using Minelute PCR purification kit from Qiagen, Germany). For single IP assay 50 µl of bead solution was used. Purified ChIPed DNA was subjected to qPCR amplification (Applied Biosystems, Carlsbad, CA). Input was used for normalization control. For primer pairs see *Supplementary file 3*.

## Antibodies

For MSL1 antibody production, a GST-mMSL1 fusion protein (C-terminal, residues 254–616) was used to immunize rabbits; the final bleed was used in experiments. Antibody specificity was verified with IP and MSL1-specific RNAi followed by Western blot analysis and ChIP assay. We used several commercial antibodies: a-KANSL1 (PAB20355; Abnova, Taiwan), a-KANSL3 (HPA035018; Sigma), a-MCRS1 (11362-1-AP; Proteintech, Chicago, IL), a-MOF (A3000992A; BETHYL Montgomery, TX), a-MSL2 (HPA003413; Sigma), a-NANOG (A300-397A; BETHYL), a-OCT3/4 (sc-5279; Santa-Cruz Dallas, TX), a-REX1 (Ab28141; Abcam, England), a-ESRRB (PP-H6705-00; Perseus Proteomics, Japan), a-KLF4 (Ab72543; Abcam), a-SOX2 (AF2018; R&D Systems, Minneapolis, MN), a-YY1 (A302-779A; BETHYL), a-RNF12/RILM (16121-1-AP; Proteintech) a-GAPDH (A300-639A; BETHYL), a-NESTIN (Ab93666; Abcam), a-CTCF (Ab70303; Abcam), a-H3 (Ab1791; Abcam), a-H4 (Ab10158; Abcam), a-H4K16ac (07-329; Millipore, Billerica, MA).

## Luciferase assays

Enhancer candidate regions (see below) were cloned into the firefly luciferase plasmids (pGL4.23; Promega, Witchburg, WI) and transfected into mouse ESCs and 3T3 fibroblasts using Lipofectamine-2000 reagent and into NPCs using LTX-PLUS reagent (Invitrogen). Transfections were performed according to the manufacturer's guidelines except for using a 1:6 DNA to Lipofectamine ratio. Cells were seeded 1 day prior to transfection to achieve 70–80% confluency at the time of transfection. Next, cells were fed with antibiotics-free medium (ES medium with LIF for ESCs and OPTIMEM for NPCs and 3T3s) at least 30 min before transfection and the medium was changed back 6–8 hr after transfection (basal neural medium with FGF and EGF for NPCs). 100 ng of firefly construct with the cloned candidate region was co-transfected with 1 ng of renilla luciferase construct (pRL-TK of Promega) per 96-well and harvested for luciferase assay after 24 hr. Cells were harvested for luciferase assay 24 hr after transfection. The Dual Luciferase Kit (Promega) was used according to the manufacturer's protocol but with reduced substrate volumes of LARII and Stop&Glo reagents (50 µl per well of a 96-well plate with 10 µl cell lysate). Luminescence was measured by using Mithras plate reader (Berthold, Germany).

The transfection efficiency was normalized by firefly counts divided by the renilla counts. The fold enhancement value was calculated by an additional normalization to minimal promoter alone activities in each experiment (the graphs represent at least three independent experiments that were performed in technical triplicates each with error bars representing standard error of the mean). The following enhancer candidate regions were amplified from mouse genomic DNA by PCR and cloned into BamHI-SalI sites (downstream of luciferase gene) of firefly luciferase plasmid pGL4.23 (Promega):

Intron of *Esrrb* (chr12:87,842,537-87,843,719) with primers introducing BamHI and XhoI sites: ATAGGATCCGAAGTAATTGTCTATTGTATCAG (forward), TATCTCGAGAAGAAGAAAGACTGTGTTCAACTCC (reverse).

Upstream of *Lefty* (chr1: 182854617-182855516) with primers introducing BamHI and SalI sites: ATAGGATCCCTTGCGGGGGATATGAGGC (forward), TATGTCGACCTGGGCCTTTCTAAGGC (reverse).

Upstream of *Trim28* (*Kap1*) (chr18: 34309039-34310140) with primer introducing BamHI and SalI sites: ATAGGATCCGAGGACTATTTGAAGGATCTATT (forward), TATGTCGACCTCACTCCCCAACCTCCATTTC (reverse).

Upstream of *Apc* (chr18: 34309039-34310140) with primers introducing BamHI and SalI sites: ATAGGATCCCTGAGCAATGCTCTTCCTCACAAGC (forward), TATGTCGACTTATACTCCAAATAGAATTGTCTG (reverse).

Intron of *Tbx3* (chr5: 120129690- 120130617) with primers introducing BamHI and SalI sites: ATAGGATCCATAAATAAATAAATAAATATCTGATTG (forward), TATGTCGACCGCGAGTCTGGCGATGCCTTGTC (reverse).

## RNA extraction followed by cDNA synthesis and quantitative real time PCR

cDNA was synthesized from 500 ng–1 µg of total RNA (extracted from circa 1 million cells using Rneasy kit, Qiagen) with random hexamers using SuperScript-III First Strand Synthesis kit (Invitrogen). The qPCRs were carried in a total reaction volume of 25 µl containing 0.5–1 µl of cDNA, 0.4 µmol of forward and reverse primer mix and 50% 2 × SYBR Green PCR Master Mix (Roche). Gene expression was normalized to multiple controls (*RplP0* or *Hprt*), using the 7500 software V2.0.4 for analysis (Applied Biosystems). For primer pairs used for expression profiling see *Supplementary file 3C*.

## Lentiviral-based RNAi in ESCs

shRNA constructs were either obtained from Sigma in pLKO.1 or designed using Genscript and cloned (please see below for details). For cloning, forward and reverse complimentary DNA oligonucleotides (Eurofins MWG Operon, Germany) designed to produce AgeI (5′) and EcoRI (3′) overhangs were annealed at a final concentration of 2 µM in NEBuffer. The pLKO.1-puro plasmid was digested with AgeI and EcoRI, ligated to the annealed oligonucleotides, and transformed into HB101 competent cells (Promega). Plasmid DNA was purified using the QIAprep Spin Miniprep kit (Qiagen), and the sequence was validated.

For production of lentiviral particles, 70% confluent HEK293FT cells in a 10-cm tissue culture plate were co-transfected with 3.33 µg lentiviral construct, 2.5 µg psPAX2 packaging plasmid and 1 µg pMD2.G envelope plasmid using Lipofectamine-2000 reagent (Invitrogen). To transduce ESCs, either

concentrated or diluted lentiviral particles were used. For concentrated lentivirus, transfections were scaled up and OPTIMEM (Invitrogen) added to the HEK293FT cells following transfection and the lentiviral supernatant collected at 48 and 72 hr post-infection. This was then concentrated using Amicon Ultra-15 centrifugal filter units (Millipore) and added to ESC media supplemented with LIF and 10 µg/ml polybrene (Millipore). For diluted lentivirus, ESC media without LIF was added to the HEK293FT cells and the lentiviral supernatant was collected after 48 hr, filtered through 0.22 µm filters (Whatmann), and added 1:1 with fresh ESC media supplemented with LIF and polybrene to the ESCs. ESCs were then subjected to selection with 1.0 µg/ml puromycin, passaged once, and harvested on day 3, 4, 5 or 6 of knockdown depending on the experiment (the numbers of days are indicated in the corresponding results section).

The following shRNA sequences were used for the knockdowns:

CCGGCCTAAGCACTCTCCCATTAAACTCGAGTTTAATGGGAGAGTGCTTAGGTTTTTG (sh*Msl1*, SIGMA, TRCN0000241378),
CCGGCCCAGTCTCTTAGCCATAATGCTCGAGCATTATGGCTAAGAGACTGGGTTTTTG (sh*Msl2*, SIGMA, TRCN0000243429),
CCGGAAGGCCGAGAAGAATTCTATCTCGAGATAGAATTCTTCTCGGCCTTTTTTTG (sh*Mof*, GENSCRIPT designed),
CCGGCTCCAGTCCTCTTCGTCATTGCTCGAGCAATGACGAAGAGGACTGGAGTTTTG (sh*Kansl3*, SIGMA, TRCN0000266995),
CCGGAAGTGGCGCCTTAGCAACAACCTCGAGGTTGTTGCTAAGGCGCACTTTTTTTG (sh*Mcrs1*, GENSCRIPT designed),
CCGGCAACAAGATGAAGAGCACCAACTCGAGACAATTCGGAAGAAATCTGAGCTTTTG (Non-targeting control, SIGMA, SHC002).

## Cell proliferation assay

Cells treated with respective shRNAs and scramble control were performed as described earlier in feeder-free W26 mouse ESCs. The cell count was monitored for 6 days post knockdown at 24-hr intervals. In brief, after 4 days of knockdown six sets of $0.4 \times 10^4$ cells per well were seeded in triplicates in a 12-well gelatinized plate. The cells were grown in ES cell culture medium supplemented with 2000 U/ml LIF and 1 µg/ml puromycin; the medium was changed every 24 hr. For counting, cells were trypsinized and counted using the Neubauer hemocytometer.

## Alkaline phosphatase staining

Detection of alkaline phosphatase, a surface marker and indicator of undifferentiated ESCs, was performed using the following method: feeder-free W26 ESCs were transduced (4 days) with scramble or the shRNAs against the genes of interest. Cells were washed twice with PBS followed by fixation with 4% PFA for 2–3 min. The cells were washed twice with PBS and stained for 20 min with staining solution (25 mM Tris-Maleic acid buffer pH 9.0, 0.4 mg/ml α-Naphthyl Phosphate (Sigma), 1 mg/ml Fast Red TR Salt (Sigma), 8 mM MgCl₂, 0.01% Deoxycholate, 0.02% NP40). The reaction was stopped by washing with water followed by two washes with 1 × PBS.

## RNA extraction for RNA-seq

Total RNA was extracted from WT26 ESCs and NPCs as biological triplicates using TRIzol Reagent and treated with the TURBO Dnase kit (Ambion).

For RNA-seq of knockdowns, feeder-free WT26 ESCs were transduced with shRNAs specific for *Msl1*, *Msl2*, *Mof*, *Kansl3* and control shRNA as biological triplicates as described above. Briefly, following transduction for 24 hr, cells were washed with PBS thrice to remove the viral supernatant and subjected to puromycin selection (1.5 µg/ml) for 24 hr. In the case of *Msl1/2*, *Mof*, control shRNA the cells were maintained in puromycin selection for 4 days and in case of *Kansl3*, the cells were maintained in puromycin-selection for 84 hr. An additional set of control shRNA was performed alongside with *Kansl3* for 84 hr. Total RNA from all the shRNA-treated cells was extracted using TRIzol Reagent and the samples were treated with DNase using the TURBO DNase kit (Ambion). The quality of the RNA was analyzed using the Bioanalyzer and samples with RIN values between 9 and 10 were used for RNA-seq. For RNA-seq analysis, cDNA libraries were prepared using the Illumina TruSeq Stranded mRNA kit with 3 µg DNase-treated samples.

## RNA-FISH

*Xist* and *Huwe1* probes were described previously (*Chow et al., 2010*). *Tsix* was detected with a *DXPas34* plasmid (*Debrand et al., 1999*). Approximately $1 \times 10^5$ of F1-21.6 ESCs were plated on gelatin-coated coverslips and incubated for 24 to 48 hr. After fixation and permeabilization, coverslips with cells were washed and stored in 70% EtOH at −20°C. Then the coverslips were dehydrated in 80%, 95%, and 100% EtOH (5 min each) and briefly air-dried. FISH probes were labeled by nick translation (Abbott) with Spectrum Red-dUTP or Spectrum Green-dUTP following the manufacturer's instructions. Labeled probes were precipitated in the presence of salmon sperm (10 µg) and Cot-1 DNA (3 µg), denatured and competed with Cot-1 DNA for 45 min at 37°C. Cells were then directly hybridized with labeled probes at 37°C overnight. Next, coverslips were washed three times in 50% formamide/2 × SSC followed by three washes in 2 × SSC at 42°C. Cells were stained with DAPI (0.2 mg/ml).

## Immunofluorescence staining (against NESTIN)

Approximately $1 \times 10^5$ of male W26 ESCs and NPCs were plated on gelatin-coated coverslips and incubated for 24 hr. The cells were washed twice with PBS and fixed with pre-warmed 4% formaldehyde for 8 min at 37°C. Next, cells were washed thrice with PBS, 5 min each at room temperature and incubated in Permeabilization buffer (1 × PBS, 0.2% Triton X-100) for 5 min at room temperature. After permeabilization cells were incubated in Blocking buffer (1 × PBS, 5% BSA, 0.05% Triton-X100) for 30 min, stained for 1 hr with primary antibody (rabbit polyclonal a-NESTIN, 1:500). Next, cells were washed thrice with Wash buffer (1 × PBS, 0.05% Triton-X100) and incubated in 10% goat normal serum solution (Invitrogen) for 20 min. Secondary antibody (goat anti-rabbit Alexa Fluor-488, 1:1000) was added on coverslips and incubated for 45 min.

## Microscopy

We used a spinning disk confocal microscope (Observer 1/Zeiss) with Plan Apochromat 63x1.4-oil objective for magnification. 500 ms exposure time was used for all lasers. Sequential z-axis images were collected in 0.5 µm steps. ZEN Blue software was used for image analysis.

## Sequencing

All samples were sequenced by the Deep Sequencing Unit (MPI-IE, Freiburg) using Illumina HiSeq2000. Library preparation was carried out following Illumina standard protocols for paired-end sequencing (50 bp reads). All raw reads can be found in the GEO database under the accession number GSE51746.

## RNA-seq data processing

RNA-seq reads were mapped to Ensembl annotation NCBIM37/mm9 using TopHat2 (*Kim et al., 2013*) with the options mate-inner-dist, mate-std-dev and library-type (fr-firststrand). The distance between read mates (mate-inner-dist and mate-std-dev) were assessed individually for each sequenced library based on the output of the sequencer for average fragment size and CV value.

For FPKM value generation, cufflinks (version 2.1.1) was used for each transcript in each condition (three replicates for ESC and NPC) with default parameters; CummeRbund was used for quality checks and data access (*Trapnell et al., 2013*). Based on the distribution of FPKM values, active genes were defined as transcripts with mean FPKM ≥4 (average over the replicates).

## Differential gene expression analysis

After mapping of the RNA-seq reads from the shRNA-treated samples (including scrambled control), the reads that mapped to the genome were counted using htseq-count (doi: 10.1101/002824) with the stranded option set to reverse. The annotations present in the *Mus musculus* gtf file from the ENSEMBL release 67 were used as reference for counting.

DESeq2 was used for differential expression analysis (*Anders and Huber, 2010*). In this analysis, all libraries from knockdown cells were compared in a pairwise manner with its corresponding scrambled shRNA samples. Within the DESeq2 workflow, the cooks–cutoff parameter was set to 'FALSE' and the genes with an adjusted p-value ≤0.01 were defined as significantly affected.

## ChIP-seq analysis

### Read mapping and normalizations

After mapping of the paired-end reads to the mouse genome (mm9) using bowtie version 2 (*Langmead and Salzberg, 2012*), we filtered for duplicate reads, reads with mapping qualities smaller than 2 and

ambiguously mapped reads using samtools (*Li et al., 2009*). We also removed reads mapping to the mitochondrial genome and 'random' chromosomes, as well as known major satellites and duplicated genome regions to avoid coverage biases.

For normalization procedures, several modules of the deepTools suite (https://deeptools.github.io) were used (*Ramirez et al., 2014*). To ensure a fair comparison between all data sets, first, the GC bias of all mapped reads was determined and, if necessary, corrected so that input and ChIP samples had similar GC distributions of their reads (correctGCbias module). In addition, all aligned read files were corrected for sequencing depth using the signal extraction method proposed by *Diaz et al. (2012)* and normalized to the cell-type-specific input (bamCompare module).

## Peak calling and replicate handling

MACS (version 1.4) was used for peak calling on every sample individually and on the merged files of two replicates (*Zhang et al., 2008*). Only peaks present in both replicates were considered, using the borders and summits defined by peak calling results for the merged replicates. In addition, peaks with $-10\log_{10}$(p-values) lower than 50 and false-discovery rate values greater than 0.1% were excluded from down-stream analyses.

## Annotation used for genome-wide analyses

We used the RefSeq gene list for genome version mm9/NCBI37. Unless specified otherwise, alternative transcription start sites were scored as individual TSS in the respective analyses. The list of genes with homologues in different species was downloaded from HomoloGene and subsequently filtered for pairs of mouse and fly genes that belong to the same clusters of homology ID. CpG island information was downloaded from the UCSC Genome Browser (*Wu et al., 2010*), mean observed over expected CpG ratios were extracted for the TSSs ± 0.5 kb using UCSC tools.

## Clustering

For *Figure 2*, a matrix containing the normalized ChIP-seq signals for all peaks was generated as follows: first, the union of peaks was created using mergeBed from the bedtools suite (*Quinlan and Hall, 2010*); then each region was binned to 2 kb and the normalized ChIP values were extracted in 50 bp windows. The ChIP signal values were rank-transformed, converted into euclidean distances using the R function 'dist' and subsequently ordered according to their similarity by the 'hclust' function (using Ward's method). The resulting dendrogram was pruned to 2 to 10 clusters for which the individual ChIP signals for unscaled regions were extracted (*Figure 2*). Visual inspection revealed no striking differences of the binding patterns between the individual clusters for more than 5 clusters.

The 3 clusters displayed in the lower part of *Figure 4—figure supplement 2B* were obtained similarly: first, a matrix was generated that contained the normalized ChIP-seq values of MOF, p300, H3K4me1 and DNase hypersensitivity sites for all regions of cluster D that did not overlap with ESC enhancers. The regions were then scaled to 1400 bp and mean values were computed for 50 bp bins using the computeMatrix module of deepTools (*Ramirez et al., 2014*). Further processing was done as described above; the resulting dendrogram was pruned to $k = 3$ and the enrichments of the different factors were computed and visualized for 10 kb regions using the heatmapper module of deepTools.

## GO term analysis

For GO term analyses, we used two approaches: the web interface of DAVID (*Huang da et al., 2009*) and GREAT (*McLean et al., 2010*).

For DAVID, we determined genes overlapping with the peaks of the individual ChIP-seq samples (TSS region ± 500 bp) and supplied the corresponding RefSeq-IDs. The background list contained the union of all TSSs bound by at least one ChIPed protein. We used the Functional Annotation Clustering tool, filtered with the option 'high stringency' and manually grouped the returned clusters of gene functions with enrichment scores above 1.3 into even broader terms.

To assess the GO terms of genes that might be regulated by the TSS-distal binding sites of MOF, MSL1, MSL2, KANSL3 and MCRS1, we used GREAT (*McLean et al., 2010*) with the mouse genome as the background data set and default settings. We obtained the top-ranked biological processes of the genes suggested to be *cis*-regulated by the regions combined in cluster D (*Figure 2*).

## Analysis of transcription factor binding sites

For the analysis of enriched transcription factor binding sites, we used the R package ChIPEnrich (http://sartorlab.ccmb.med.umich.edu/chip-enrich) and TRAP (*Thomas-Chollier et al., 2011*). The ChIPEnrich package takes peak regions as input and uses a logistic regression approach to test for gene set enrichments while normalizing for mappability and locus length. We supplied the regions belonging to the individual clusters of binding (A–E from *Figure 2*) and obtained the corresponding enriched transcription factors.

To plot the occurrences of the SMAD3 motif (V$SMAD3_Q6, TRANSFAC name M00701; *Figure 4— figure supplement 3C*), TRAP was used with the following command to generate a bedgraph file where the log likelihood of a SMAD3 motif occurrence is stored for the entire genome: ANNOTATEv3.04_source/Release/ANNOTATE_v3.04 -s mm9.fa --pssm /transfac.pssm -g 0.5 --ttype balanced -name M00701 -d | awk 'BEGIN{OFS=t}{print $1, $4+7, $4+8, $6}' > SMAD3. pssm.bedgraph.

## Heatmap visualizations and summary plots

Heatmaps displaying normalized read densities of ChIP-seq samples, % methylated CpGs and SMAD3 motif score (*Figures 2, 3C, 4A*, *Figure 4—figure supplement 2B and 3*) were generated with the computeMatrix and heatmapper modules of the deepTools package (*Ramirez et al., 2014*) with 'reference-point' mode. Heatmaps of fractions of overlapped regions as in *Figure 3— figure supplement 1* and *Figure 4B* as well as $\log_2$ fold changes (knockdown/control) from RNA-seq experiments (*Figure 3E*) were generated with the function 'heatmap.2' from the R gplots package.

The values underlying the summary plots such as the meta-gene and meta-enhancer plots in *Figures 3D and 4B*, *Figure 3—figure supplement 2B*, *Figure 4—figure supplement 1B, 2A,C* were generated with the computeMatrix module of the deepTools package using either 'reference-point' or 'scale-regions' mode and were visualized with the R package ggplots2.

## Working with genomic intervals

For general assessments of overlaps between bed-files and to extract scores for defined regions the bedTools suite (*Quinlan and Hall, 2010*) and UCSC tools (*Kuhn et al., 2013*) were used. The snapshots of the binding profiles were obtained with IGV browser (*Thorvaldsdottir et al., 2013*).

## Target definitions

For each knockdown condition for which RNA-seq data had been generated (see above), significantly affected genes were used (adjusted p-value ≤0.01, see above for differential gene expression analysis). Then they were subdivided into TSS- (ChIP-seq peak overlap with TSS ±1 kb), TSS-distal- (ChIP-seq peaks not overlapping with TSS ±1 kb) and non-targets (neither TSS overlap nor part of TSS-distal list). A gene was classified as TSS-distally regulated when at least one of the following criteria was true:

1. TSS-distal peaks overlapped its published super or typical enhancer (*Whyte et al., 2013*)
2. TSS-distal peaks were predicted by GREAT (*McLean et al., 2010*) to regulate the respective gene
3. TSS-distal peaks overlapped with at least one intron

Genes were defined as MSL targets when peaks of MOF and MSL1|MSL2 were overlapping at the TSS ±1 kb or TSS-distal peaks were predicted to regulate the same putative target gene. NSL targets were defined the same way, but with co-occurrences of peaks from MOF and KANSL3|MCRS1.

## Acknowledgements

We would like to thank T Jenuwein from the MPI-IE, Freiburg for providing the WT26 ES cell line and U. Riehle from the Deep Sequencing Unit of the MPI-IE, Freiburg for sequencing all samples. We would also like to thank P Kindle (MPI-IE, Freiburg, Imaging facility), S Toscano, and M. Shvedunova for help with imaging. TC is especially grateful to A Chatterjee for providing support and insightful discussions. We thank all members of the labs for helpful discussions. This work was supported by DFG-SFB992 and BIOSSII awarded to AA and RB; DFG-SFB746 awarded to AA. AA is also a member of the EU-NoE-EpiGeneSys.

## Additional information

### Competing interests

AA: Reviewing editor, *eLife*. The other authors declare that no competing interests exist.

### Funding

| Funder | Grant reference number | Author |
|---|---|---|
| SFB DFG Germany | SFB992 | Rolf Backofen, Asifa Akhtar |
| SFB DFG Germany | SFB746 | Asifa Akhtar |
| DFG Germany | BIOSS II | Rolf Backofen, Asifa Akhtar |
| EU NoE | EpiGeneSys | Asifa Akhtar |

The funders had no role in study design, data collection and interpretation, or the decision to submit the work for publication.

### Author contributions

TC, FD, Conception and design, Acquisition of data, Analysis and interpretation of data, Drafting or revising the article; MT, TK, TA, FR, Acquisition of data, Analysis and interpretation of data; A-VG, Acquisition of data, Contributed unpublished essential data or reagents; PRW, PV, RB, TM, Analysis and interpretation of data, Drafting or revising the article; EH, Drafting or revising the article, Contributed unpublished essential data or reagents; AA, Conception and design, Analysis and interpretation of data, Drafting or revising the article, Contributed unpublished essential data or reagents

## Additional files

### Supplementary files

• Supplementary file 1. MOF/MSL/NSL ChIP-seq statistics.

• Supplementary file 2. Detailed information about publicly available genome-wide resources used in this study.

• Supplementary file 3. Lists of primer pairs used in ChIP-qPCR and RT-PCR analyses.

### Major dataset

The following dataset was generated:

| Author(s) | Year | Dataset title | Dataset ID and/or URL | Database, license, and accessibility information |
|---|---|---|---|---|
| Duendar F | 2013 | MOF-associated complexes ensure stem cell identity and Xist repression | http://www.ncbi.nlm.nih.gov/geo/query/acc.cgi?acc=GSE57701 | Publicly available at NCBI GEO. |

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
