## [Decision Letter]

Thank you for sending your work entitled “MOF complexes use short and long-range interactions to ensure stem cell identity and Xist repression” for consideration at *eLife*. Your article has been evaluated by a Senior editor, a Reviewing editor, and 3 reviewers, one of whom, Barbara Panning, has agreed to reveal her identity.

The Reviewing editor and the reviewers discussed their comments before we reached this decision, and the Reviewing editor has assembled the following comments to help you prepare a revised submission.

The reviewers were generally supportive of the content of the manuscript and felt that it added new important information on the potential roles of the MOF complexes in stem and progenitor cells. However, there were a number of queries about the interpretation of the results and suggestions for more experimental data that would be needed to make the paper acceptable for publication in *eLife*. If you feel you can address all of the issues and provide the appropriate experimental data, we will be happy to consider a revised manuscript.

It would seem that some more experiments on the biological effects of depletion of the complexes should be included. The phenotype of ES cells after depletion of NSL and MSL proteins such as ES proliferation and differentiation need to be included for documenting the biological function and relevance. The properties of the NPC cells should be better described, along with the efficiency of the differentiation process (i.e., what proportion of derived cells are definitive NPC). It would also be interesting to have comments on the positioning of the MSL/H4K16ac peak across the DXPas34 region of the X. This is some distance from the Tsix promoter. What does this rather precise positioning tell us about MSL's function in this region?

From the presented data it appears highly unlikely that the individual M/NSL proteins act through a coherent mechanism. The text often refers to “Both complexes bind to ...” but this might be misleading as there is hardly any binding of “complexes” rather than individual proteins acting independently. This weakens the manuscript as interesting data that Msl2 and Smads might be interacting (Figure 4; is this independent of Mof?). Our impression when reading the paper was that a lot of speculative correlations are included where no biological function can be clearly associated which may be exhausting to some readers. We suggest you focus the study on biologically relevant observations and demonstrate their functional importance. Does Msl2 depletion affect Smad signaling?

---

## [Author Response]

*It would seem that some more experiments on the biological effects of depletion of the complexes should be included*.

We thank the reviewers for this point. Upon revision we have now added a series of new experiments to further elaborate on the biological significance of the MSL and NSL proteins. In short, we added the following findings:

We show the overall effects on ESC growth and differentiation upon MSL1, MSL2, MOF and KANSL3 depletion. We added genome-wide RNA-seq analyses in MSL1-, MSL2-, MOF- and KANSL3-depleted ES cells. We added additional Xist RNA-FISH experiments upon ESC differentiation to show the dramatic acceleration of Xist-mediated X-chromosomal coating leading to X inactivation and upon MSL1 and MSL2 depletion. We also observe ‘chaotic’ coating of both X chromosomes previously associated with Tsix-DXPas34 deletion, which has been implicated in the processes of counting and choice (40, 77).

We strongly believe that the new datasets added upon revision not only elucidate the biological significance of these complexes in mammals in more detail; they also highlight a beautiful analogy of the MSL-specific function in Drosophila and mammals to keep the X chromosome active.

*The phenotype of ES cells after depletion of NSL and MSL proteins such as ES proliferation and differentiation need to be included for documenting the biological function and relevance*.

We thank the reviewers for this comment to strengthen the manuscript. We have now added the following newly generated data to provide insights into the biological function and relevance of the NSL and MSL proteins.

Growth and morphological analyses in ES cells: In the figure supplements to Figures 3 and 4, we now added growth curves and cell morphology data including AP staining upon MSL1, MSL2, KANSL3 and MOF depletion in ES cells.

Genome-wide RNA-seq analyses in MSL1-, MSL2-, MOF- and KANSL3-depleted ES cells: To obtain a global overview of the extent of mis-regulation following depletion of MSL and NSL proteins, we performed thorough RNA-seq analyses of control, MSL1-, MSL2-, KANSL3- and MOF-depleted male ES cells. The results provide novel insights into the biological function of these proteins. We find that MOF-and KANSL3-depletion particularly affect genes that were determined as TSS targets in the ChIP-seq analysis while MSL1 and MSL2 predominantly affect TSS-distal target genes. This data further demonstrates the distinct functions of these proteins in ES cells.

*The properties of the NPC cells should be better described, along with the efficiency of the differentiation process (ie. what proportion of derived cells are definitive NPC)*.

We used a well-established 4-stage procedure that has been shown before to give rise to homogeneous populations of NPCs (11, 65). The efficiency of this method has been proven in the field in many occasions (22). In addition, we selected for NPCs growing in monolayers and exhibiting identical morphology by keeping cells for 5-10 passages to filter out the remaining embryoid bodies. To support our results, we now include an overview of expression values for ESC- and NPC-markers as well as immunostaining for NESTIN (well characterized marker of NPCs) in ESCs as well as NPCs in Figure 1—figure supplement 1.

*It would also be interesting to have comments on the positioning of the MSL/H4K16ac peak across the DXPas34 region of the X. This is some distance from the Tsix promoter. What does this rather precise positioning tell us about MSL's function in this region*?

We thank the reviewers for this important point. We added a brief paragraph discussing this in the text. We would also like to draw the reviewers’ attention to the fact that Tsix possesses two promoters and that the major promoter (to which we refer in the manuscript as P2) is in close proximity to the DXPas34 locus. It has been also previously shown that the DXPas34 locus on its own (without promoter region) can support the transcription of Tsix and it acts as the major enhancer for Tsix}. We now specified the exact location of MSL1, MSL2 and MOF across the given locus in the text as well as in the respective figures and extended information about properties and importance of Tsix and DXPase34 locus in the last section of results (Figure 7) and discussion.

“The XIC site with the strongest concomitant enrichments of MSL1, MSL2 and MOF was the major promoter (P2) of Tsix and its intronic minisatellite – DXPas34 (Figure 5). DXPas34 is a well-characterized tandem repeat that serves as a binding platform for multiple transcription factors and contains bidirectional enhancing properties essential for the expression of Tsix, the antisense transcript of Xist (14, 9, 19, 51, 23). In rodents, Tsix antisense transcription across the Xist promoter is required for regulating the levels of Xist accumulation. In turn, DXPas34 deletion impairs the recruitment of Pol II and TFIIB to the major promoter of Tsix causing its downregulation (77).”

*From the presented data it appears highly unlikely that the individual M/NSL proteins act through a coherent mechanism. The text often refers to “Both complexes bind to ...” but this might be misleading as there is hardly any binding of “complexes” rather than individual proteins acting independently. This weakens the manuscript as interesting data that Msl2 and Smads might be interacting (*Figure 4*; is this independent of Mof?)*.

We agree with the point raised by the reviewers. Indeed, our data shows that MSL and NSL proteins could work synergistically but also independently of other complex members in vivo. We have revised our text and analyses and have clearly stated when we refer to the MOF-containing complexes (i.e., MSL = MOF + MSL1 + MSL2; NSL = MOF + KANSL3 + MCRS1) or the individual complex members.

Just like the reviewers, we are also intrigued by the distinct MSL2 binding and this finding definitely merits an independent and thorough analysis which is beyond the scope of this manuscript. We believe that this MOF-independent MSL2 binding is an important piece of information for the epigenetics community, which we would like to keep in the manuscript. As, indeed, we cannot offer much more than speculations on the functional implication of these binding sites, we have moved Figure 4 to the supplementary information.

*Our impression when reading the paper was that a lot of speculative correlations are included where no biological function can be clearly associated which may be exhausting to some readers. We suggest you focus the study on biologically relevant observations and demonstrate their functional importance*. *Does Msl2 depletion affect Smad signaling?*

We have now modified the text so that the emphasis is given on targets where we observe an effect upon depletion of the ChIPed proteins. We have also expanded and added additional data on the regulation of Tsix to further strengthen the direct role of the MSL complex in regulation of Tsix. Eventhough we have extended our results section (due to the addition of new data) we have shortened our discussion to focus on those observations directly supported by our experiments.

For MSL2/Smad connection, please see the response above.